# **Evaluation of Wet Snow Dielectric Mixing Models for L-Band Radiometric Measurement of Liquid Water Content in Greenland's Percolation Zone**

Alamgir Hossan<sup>1</sup>, Andreas Colliander<sup>1</sup>, Nicole-Jeanne Schlegel<sup>2</sup>, Joel Harper<sup>3</sup>, Lauren Andrews<sup>4</sup>, Jana Kolassa<sup>4,5</sup>, Julie Z Miller<sup>6</sup>, Richard Cullather<sup>4,7</sup>

<sup>1</sup>Jet Propulsion Laboratory, California Institute of Technology, Pasadena, California, United States

<sup>2</sup>NOAA/OAR Geophysical Fluid Dynamics Laboratory (GFDL), Princeton, New Jersey, United States

<sup>3</sup>Department of Geosciences, University of Montana, Missoula, Montana, United States

<sup>4</sup>NASA GSFC/GMAO Greenbelt,- Maryland, United States MD

<sup>5</sup>SSAI, Berwyn Heights, Maryland, United States MD Department of Geography, University of Calgary

<sup>6</sup>Cooperative Institute for Research in Environmental Sciences, University of Colorado Boulder, <u>Boulder, Colorado, United</u>
States

<sup>7</sup>Univ. of Maryland at College Park, ESSIC, College Park, Maryland, United States MD

Correspondence to: Alamgir Hossan (<u>alamgir.hossan@jpl.nasa.gov</u>) and Andreas Colliander 20 (andreas.colliander@jpl.nasa.gov)

Abstract. Determining the effective permittivity of snow and firm is essential for the accurate estimation of liquid water amount (LWA). Here, we compare ten commonly used microwave dielectric mixing models for estimating LWA in wet snow and firm using L-band radiometry. We specifically focus on the percolation zone of the Greenland Ice Sheet (GrIS), where the average volume fraction of liquid water is approximately 6 percent. We used L-band brightness temperature (TB) observations from the NASA Soil Moisture Active Passive (SMAP) mission in an inversion-based framework to estimate LWA, applying different dielectric mixing formulations in forward simulation. We compared the effective permittivities of the mixing models over a range of conditions and evaluated their impacts on the LWA retrieval. We also compared the LWA retrievals to the corresponding LWA from two state-of-the-art Surface Energy and Mass Balance (SEMB) models. Both SEMB models were forced with in situ measurements from automatic weather stations (AWS) of the Programme for Monitoring of the Greenland Ice Sheet (PROMICE) and Greenland Climate Network (GC-Net) located in the percolation zone of the GrIS and initialized with relevant in situ profiles of density, stratigraphy, and sub-surface temperature measurements. The results show that the mixing models produce substantially different real and imaginary parts of the dielectric constant. The choice of mixing model has a significantly impacting on the LWA retrieved from the TB. The correspondence with the SEMB-derived LWA varied by model and site. With correlation coefficients ranging from 0.67 to 0.98 and RMSD values between 5.4 and 23.9 mm. Overall, the power law-based models demonstrated better performance the Sihvola power law based mixing model showed an

overall better performance than the other models for 2023 melt season. The analysis facilitates supports an appropriate informed choice selection of dielectric mixing models on the for improved LWA retrieval algorithmaccuracy.

# 1 Introduction

55

Surface melting and consequent runoff/refreezing play an increasingly major role in the Greenland Ice Sheet (GrIS) surface mass balance (SMB) and its contribution to the global sea-level rise (Greene et al., 2024; Khan et al., 2022; Khan et al., 2015; Mouginot et al., 2019; Otosaka et al., 2023; Shepherd et al., 2020). The column-integrated amount of liquid water (LWA) at the surface and percolated within layers of the surface snowpack is a key variable for understanding processes related to meltwater on the ice sheet surface, and is thus an important quantity for diagnostic study, modeling, and prediction. Currently, there is no direct means of measuring LWA in the ice sheet. In situ AWS provides surface meteorological observations for limited locations over the ice sheet, which are translated to LWA estimates using coupled surface energy balance and subsurface hydrology and heat transfer models (Fausto et al., 2021; Samimi et al., 2021; Vandecrux et al., 2020). Regional climate models provide pan-ice sheet estimates of LWA (Fettweis et al., 2020), but uncertainty results from the significant differences in the configuration and physical process representations in these models (Thompson-Munson et al., 2023; Fettweis et al., 2020; Vandecrux et al., 2020; Verjans et al., 2019). Spaceborne microwave radiometers have also been used for large-scale mapping of polar ice sheet melt (Picard et al., 2022; Tedesco, 2007; Tedesco et al., 2007; Abdalati and Steffen, 1997; Mote and Anderson, 1995; Zwally and Fiegles, 1994). However, shallow penetration into the wet snow restricts the conventional high frequency radiometers (i.e., greater than 6 GHz) to providing only surface and near-surface binary melt status, and not the actual volumetric amount of liquid water in the snow/firn (Leduc-leballeur et al., 2025; Colliander et al., 2022a, b, 2023; Mousavi et al., 2022).

The higher penetration of L-band radiometry offers a promising new tool for quantifying the total surface-to-subsurface LWA in the firn, in addition to providing the areal extent and duration of seasonal surface snow melt (Houtz et al., 2019, 2021; Mousavi et al., 2021; Schwank and Naderpour, 2018; Colliander et al., 2022a; Colliander et al., 2022b; Miller et al., 2020a, 2022a, b; Mousavi et al., 2022). Houtz et al. (2019 and 2021) used L-band brightness temperature (TB) from the European Space Agency's Soil Moisture and Ocean Salinity (SMOS) mission for simultaneous estimation of snow liquid water content and density in the GrIS. They used the Microwave Emission Model of Layered Snowpacks Version 3 (MEMLS V3; Mätzler and Wiesmann, 2012) with L-band specific modifications (LS-MEMLS; Schwank et al., 2014) in an inversion-based retrieval framework. By default, MEMLS V3 uses the Mätzler (1996) and Mätzler and Wiesmann, (2007) formulations for dielectric mixing of dry and wet snow, respectively. Naderpour et al. (2021) used the same algorithm to quantify LWA at the Swiss Camp location (70°N, 49°W) with close-range (CR) single-angle L-band microwave radiometer measurements. Mousavi et al. (2021) developed an L-band specific snow/firn radiative transfer model that uses the Mätzler (2006) and Ulaby et al. (2014) dielectric mixing model for dry and wet snow, respectively, to estimate LWA. Hossan et al. (2024) used the same approach to quantify and validate the LWA with two surface energy balance models forced with in situ observations and reanalysis data

products. Additionally, Moon et al. (2024) compared the Hossan et al. (2024) retrieval with LWA values derived from subsurface thermal measurements. The study showed mixed correspondence of the L-band retrievals to the alternative LWA estimates.

The L-band TB responds to the real and imaginary parts of the firn dielectric constant, which increases markedly with volumetric liquid water content (snow wetness or the volume fraction of liquid water in the snow mixture,  $v_w$  hereafter, expressed as a percentageLWC) in the firn (Picard et al., 2022; Samimi et al., 2021; and references therein). The measured dielectric constant is translated into LWA using a model between snow  $v_w$ LWC and the dielectric constant. The formulation of the effective dielectric constant of the ice, air, and water mixture is key to accurately quantifying LWA. As it is independent of the radiometer measurement, it adds an uncertainty component to the LWA retrieval that is solely dependent on the accuracy of this dielectric mixing model. Picard et al. (2022) demonstrated large differences in commonly used wet snow dielectric mixing models for both the real and imaginary parts.

In this manuscript, we assess the performance of ten commonly used microwave dielectric mixing models in quantifying the seasonal LWA using L-band (1.4 GHz) enhanced-resolution (rSIR) TB observations data products from the Soil Moisture Active Passive (SMAP) mission. For this, we confine our attention to the GrIS percolation zone where the average volume fraction liquid water inclusions in the snow/firn environment is within about 6 percent of the total volume (Colbeck, 1974; Coléou and Lesaffre, 1998), and L-band TB during melting is mainly dominated by absorption (increasing trend compared to frozen season; Hossan et al., 2024).

## 2 Methods

85

70

Snow and firn (the transitional snow that survivsurvivesed at least a summer season) is generally a three-phase porous dielectric mixture of air, ice, and liquid water, where dry snow is a special case having no liquid water. Here we briefly discuss the dielectric properties of snow and firn.

## 2.1 Dielectric Mixing Formulas

Dielectric mixing rules attempt to approximate the effective macroscopic dielectric constant (permittivity)  $\varepsilon_{eff}$  of snow/firn mixture that relates the average electric flux density  $\boldsymbol{D}$  -to the incident or emitting electric field  $\boldsymbol{E}$  and the average polarization  $\boldsymbol{P}$  of the mixer (Sihvola, 1999; Jones and Friedman, 2000),

$$\mathbf{D} = \varepsilon_{eff} \mathbf{E} = \varepsilon_{e} \mathbf{E} + \mathbf{P} \tag{1}$$

wWhere  $\varepsilon_e$  is the complex dielectric constant of the background medium. The polarization term, P, represents the number of dipole moments per unit volume in the mixture and is a function of inclusion geometry.  $\varepsilon_{eff}$  thus, depends on the individual dielectric constant of the constituent materials, their respective volume fractions, and their size, shape, and orientations. For a

two phase mixture. The a generalized mixing formula can be derived from the Maxwell Garnett (MG) mixing rule (Garnett, 1904),

$$\varepsilon_{eff} = \varepsilon_e + 3v\varepsilon_e \frac{\varepsilon_i - \varepsilon_e}{\varepsilon_i + 2\varepsilon_e - v(\varepsilon_i - \varepsilon_e)} \tag{2}$$




where  $\varepsilon_e$  and  $\varepsilon_i$  is the arecomplex dielectric constants of the background environment (host) and inclusion (guest), respectively, and v is the volume fraction of the inclusion. However, tThe fundamental MG rule considers only dilute concentration of spherical inclusions ( $v \ll 1$ ) (Jones and Friedman, 2000) and assumes homogeneous inclusionsmixtures, ignoring the second-order effects due to the mutual interactions between the inclusions. When the inclusions are arbitrarily spread within the host material, the fields within the inclusions are a function of the mutual interactions of the inclusions (through their polarization fields P). The interactions among the inclusions are dependent on their relative geometry and alignment, which are both taken into account by a parameter called the depolarization factor. For an ellipsoid, the depolarization factor along its u axis is given by (Jones and Friedman, 2000; Sihivola, 1999),

$$N_{u} = \frac{abc}{2} \int_{0}^{\infty} \frac{ds}{(s+u^{2})\sqrt{(s+a^{2})(s+b^{2})(s+c^{2})}} \quad u = a, b, c$$
 (3)

where  $N_a + N_b + N_c = 1$ . Aspect ratios of the axial dimensions (u) of the particle describe the shape of the particle, a: b = 1 defines the spherical inclusions whereas a: b < 1 and a: b > 1 describe oblate and prolate inclusions, respectively. The depolarization factor  $N_u$  is commonly included in the more general mixing formula described below.

The Bruggeman mixing rule (Long and Ulaby, 2015; Sihvola, 1999) considers mixing phases to be symmetric to describe the effective permittivity of a mixture as an implicit function of unknown effective permittivity of the mixture, which for the case of randomly oriented ellipsoidal inclusions (Sihvola, 1999) reads as,

$$\varepsilon_{eff} = \varepsilon_e + \frac{v}{3}(\varepsilon_i - \varepsilon_e) \sum_{u=a,b,c} \left[ \frac{\varepsilon_{eff}}{\varepsilon_{eff} + N_u (\varepsilon_i - \varepsilon_{eff})} \right]$$
 (4)

Polder and van Santen (1946) (PVS hereafter) derived similar implicit formulation for a two-phase mixture with randomly oriented ellipsoidal inclusions. Loor (1968) extended the work as follows,

$$\varepsilon_{eff} = \varepsilon_e + \frac{v}{3} (\varepsilon_i - \varepsilon_e) \sum_{u=a,b,c} \left[ \frac{1}{1 + N_u \left( \frac{\varepsilon_i}{\varepsilon_*^*} - 1 \right)} \right] \tag{5}$$

where  $\varepsilon^*$  is the effective dielectric constant of the region surrounding inclusions. For  $v \leq 0.1$ ,  $\varepsilon^* \approx \varepsilon_e$  and for higher value of f,  $\varepsilon^*$  approaches to  $\varepsilon_i$  (Loor, 1968).

Another widely used mixing rule is the Coherent Potential (CP here after) formula (Tsang et al., 1985), which for the case of randomly oriented ellipsoidal inclusions is given by (Sihvola, 1999),

$$\varepsilon_{eff} = \varepsilon_e + \frac{v}{3} (\varepsilon_i - \varepsilon_e) \sum_{u=a,b,c} \left[ \frac{(1+N_u)\varepsilon_{eff} - N_u\varepsilon_e}{\varepsilon_{eff} + N_u(\varepsilon_i - \varepsilon_{eff})} \right]$$
 (6)

For dilute inclusion ( $v \ll 1$ ), all of these formulas (Eq. 4\_-6) provide the same results as the MG mixing rule (Sihvola, 1999). However, as v increases the MG formula usually predicts  $\varepsilon_{eff}$  closer to  $\varepsilon_e$  (dielectric constant of host or background environment) which is lower than that estimated by both the PVS and CP formulations. This is because, as mentioned above, MG neglects the second-order effects due to the mutual interactions between neighbouring inclusions. The CP formula considers the effective medium instead of the background to find the local field and, therefore, estimates higher  $\varepsilon_{eff}$  compared to  $\varepsilon_e$ . The PVS formula, on the other hand, represents a balance between MG and CP, as it treats both the inclusions and the surrounding environment symmetrically, resulting in an equal influence from the permittivity of the two phases (Jones and Friedman, 2000; Sihivola, 1999). However, a computational difficulty of PVS and CP formulae is that they are both implicit in  $\varepsilon_{eff}$ .






Tinga et al. (1973) derived an explicit formula for  $\varepsilon_{eff}$  by considering a two-phase mixture composing of two randomly oriented confocal ellipsoids with an inner ellipsoid representing the inclusion and an outer ellipsoidal shell representing the host material.

$$\varepsilon_{eff} = \varepsilon_e + \frac{v}{3} (\varepsilon_i - \varepsilon_e) \sum_{u=a,b,c} \left[ \frac{1}{1 + (N_{u2} - fN_{u1}) (\frac{\varepsilon_i}{\varepsilon_e} - 1)} \right]$$
 (7)

where  $N_{u2}$  and  $N_{u1}$  are the depolarization factors of the inner and outer ellipsoids, respectively. It is noted that for spherical inclusions ( $N_{u2} = N_{u1} = \frac{1}{3}$ ), Eq. 7 <u>also</u> reduces to the MG mixing rule, but any deviations from the spherical shape increase the  $\varepsilon_{eff}$  if  $\varepsilon_i > \varepsilon_e$  (and vice versa).

Another group of mixing formulas follows power-law relations, where a certain power of effective permittivity of a multi-phase mixture relates to the linear combination of components raised to the same power and weighted by their respective volume fractions,—v (Sihvola et al., 1985; Sihvola, 1999). These exponential models do not explicitly consider the microstructure shapes (i.e. through depolarization factor, N), but) but take into account the higher order mutual interactions through the power-law averaging. The general form of these models takes the form,

145 
$$\varepsilon_{eff}{}^{\beta} = \sum_{j} v_{j} \varepsilon_{j}{}^{\beta}$$
 (8)

where  $v_j$  and  $\varepsilon_j$  are the volume fraction and dielectric constant of the j<sup>th</sup> constituent, respectively, and  $\sum_j v_j = 1$ . The exponent  $\beta$  controls the degree of nonlinearity of the model (Sihvola et al., 1985), which is bounded by  $0 < \beta \le 1$ . The lower the value of  $\beta$ , the higher the influence of the background (dominant volume fraction).

The effective dielectric constant of a mixture is also calculated in the frequency domain using dispersion models. In case of inclusions with permanent electric dipole moments, like liquid water in wet snow, the Debye relaxation model is best suited (Hallikainen et al., 1986; Sihvola, 1999). Wet snow shows a distinct Debye relaxation spectrum in the microwave range (Ulaby and Long, 2014). The Debye-like semi-empirical models are of the form (Hallikainen et al., 1986),

$$\varepsilon'_{eff} = A + \frac{Bv_w^{\chi}}{1 + (\frac{f}{f_0})^2} \tag{9.1}$$

$$\varepsilon''_{eff} = \frac{c(\frac{f}{f_0})v_w^{\chi}}{1+(\frac{f}{f_0})^2} \tag{9.2}$$

where  $\varepsilon_{eff}'$  and  $\varepsilon_{eff}''$  are the real and imaginary parts of the effective dielectric constant of the mixture and  $v_w$  is the volume fraction of liquid water in snow. f and  $f_0$  are the operational and relaxation frequencies respectively and A, B, C, and x are constants that are determined empirically by fitting experimental data.

There are numerous models and formulas in the literature describing the dielectric behaviour of mixtures. Comprehensive reviews on the topic can be found in Sihvola (1999) and the references therein. Many of these formulas are special cases or modifications of the above basic mixing rules. Some others are empirical in nature. In the following section, we will briefly describe some specific wet snow mixing models that we evaluated in this study.

## 2.2 Dielectric Constant of Dry Snow

160

Dry snow is a two-phase mixture of ice and air. Since the real part of the dielectric constant of ice,  $\varepsilon'_i$ , is independent of frequency and almost independent of temperature, it is assumed that the real part of the dielectric constant of dry snow,  $\varepsilon'_{ds}$  is also independent of both frequency and temperature (Hallikainen et al., 1986).  $\varepsilon'_{ds}$  is thus fully determined by the density of the dry snow (Denoth, 1989; Denoth et al., 1984; Tiuri et al., 1984). However, the imaginary part of ice dielectric constant,  $\varepsilon''_i$ , and thus the dry snow  $\varepsilon''_{ds}$  are strongly sensitive to both frequency and temperature (Ulaby and Long, 2014). With known dielectric constants of air and ice, the above dielectric mixing models, such as two phase PVS mixing rule, can be applied to find the effective dielectric constant of dry snow. Empirical formulations based on experimental data also provide good results (e.g., Mätzler, 2006).

For the real part of dry snow permittivity, we follow the empirical relation presented in Mätzler (2006),

$$\varepsilon'_{ds} = \begin{cases} 1 + 1.4667v_i + 1.435v_i^3 & for \ 0 \le v_i \le 0.45\\ (1 + 0.4759v_i)^3 & for \ v_i \ge 0.45 \end{cases}$$
(108)

where  $v_i$  is the volume fraction of ice in snow given by the ratio of snow and ice density.

<u>FAnd for</u> the imaginary part,  $\varepsilon''_{ds}$ , we follow the Hallikainen et al., (1986) formulation based on Tinga mixing model (Eq. 7),

$$\varepsilon''_{ds} = \frac{0.34v_i\varepsilon''_i}{(1-0.42v_i)^2} \tag{119}$$

where  $\varepsilon''_{i}$  is the imaginary part of the dielectric constant of ice determined based on Mätzler (2006).

Various formulations are available for the effective dielectric constant of dry snow. Since our focus in this manuscript 180 is wet snow mixing models, we apply the same dry snow model (Eqs. 8-9) (Mätzler, 2006) in all following cases for the mixture where the host is dry snow.

#### 2.3 Dielectric Constant of Wet Snow

In this subsection, we discuss ten commonly used dielectric mixing models for estimating the complex dielectric constant of wet snow. Mätzler model (Mätzler and Wiesmann, 2012; used in MEMLS V3) uses the MG rule to compute the effective dielectric constant of wet snow as a two-phase mixture of liquid water inclusions in a dry snow background (host) with experimentally determined depolarization factors  $N_a = 0.005$ ,  $N_b = N_c = 0.4975$  (representing a prolate spheroidal shape of the inclusion) from Hallikainen et al. (1986) and Matzler et al. (1984),

$$\varepsilon_{eff} = \frac{(1 - v_w)\varepsilon_{ds} + v_w \varepsilon_w K}{(1 - v_w) + v_w K} \tag{12.1}$$

$$K = \frac{1}{3} (K_a + K_b + K_c)$$
 (12.2)

$$K_u = \frac{\varepsilon_{ds}}{\varepsilon_{ds} + N_u (\varepsilon_w - \varepsilon_{ds})} \quad u = a, b, c$$
 (12.3)

where  $\varepsilon_w$  is the dielectric constant of pure water which is determined from Liebe et al. (1991) relaxation formula. 2.3.5 Tinga Model

The Tinga model (Tinga et al., 1973) also considers wet snow as a three-phase mixture (air-snow-liquid water) where air is the background and water is a spherical shell surrounding another confocal shell (ice). The effective dielectric constant is then determined following Eq. 7.

## 2.3.1 Debve like Model

The frequency dependence of the wet snow mixture is highly influenced by the dispersion property of water (Hallikainen et al., 1986; Sihivola, 1999). It shows a distinct Debye relaxation spectrum in the microwave range (Ulaby and Long, 2014). Hence a Debye like semi-empirical model is often used to describe the polarization response of liquid water in wet snow. The models are of the form (Hallikainen et al., 1986).

$$\varepsilon'_{eff} = \Lambda + \frac{Bv_{\overline{w}}^{\pm}}{1 + (\frac{f}{f_{\theta}})^2} \tag{10.1}$$

$$\varepsilon'_{eff} = A + \frac{Bv_{w}^{*}}{1 + (\frac{f}{f_{\Phi}})^{2}}$$

$$\varepsilon''_{eff} = \frac{c(\frac{f}{f_{\Phi}})v_{w}^{*}}{1 + (\frac{f}{f_{\Phi}})^{2}}$$

$$(10.1)$$

where  $\varepsilon_{eff}$  and  $\varepsilon_{eff}$  are the real and imaginary parts of the effective dielectric constant of the mixture and where  $v_w$  is the volume fraction of liquid water in snow. f and  $f_0$  are the operational and relaxation frequencies respectively and A, B, C, and x are constants that are determined empirically by fitting experimental data. One such approach (Hallikainen et al., (1986) first proposed a Debye-like model for wet snow recommends with the following expression for the constants in Eq. 9.

$$A = 1 + 1.83 \,\rho_{ds} + 0.02 \,A_1 v_w^{1.015} + B_1 \tag{134.1}$$

$$B = 0.073 A_1 \tag{134.2}$$

$$C = 0.073 A_2 \tag{134.3}$$

$$x = 1.31$$
 (134.4)

$$f_0 = 9.07 \,\text{GHz}$$
 (134.5)

where for the original Debye-like model,  $A_1 = A_2 = 1$ , and  $B_1 = 0$ .

# 215 2.3.2 Modified Debye-like Model (Hallikainen et al., 1986)

Hallikainen et al., (1986) derived the expressions below for the constants  $A_1$ ,  $A_2$ , and  $B_1$  in Eq. 143 as a function of frequency by fitting Eq. 134 to the field measurements of volumetrie  $v_w$ LWC with a range 0 - 12 percent, a density range of 0.09 - 0.42 g cm<sup>-3</sup>, a temperature range of -15°C - 0°C, grain radius covering 0.5 - 1.5 mm at the frequency range of 3 to 37 GHz (see Hallikainen et al., 1986).

$$A_1 = 0.78 + 0.03f - 0.58 \times 10^{-3} f^2 \tag{124.1}$$

$$A_2 = 0.97 - 0.39f \times 10^{-2} + 0.39 \times 10^{-3} f^2$$
 (124.2)

$$B_1 = 0.31 - 0.05f + 0.87 \times 10^{-3}f^2 \tag{124.3}$$

where f is in GHz. Here we test the applicability of this model for L-band and full possible density range in the percolation zone of the GrIS. For f = 1.4 GHz (L-band), the values of  $A_1$ ,  $A_2$ , and  $B_1$  are 0.82, 0.96, and 0.24 respectively. Hereafter, we refer to this model as Hallikainen model for simplicity.

## 2.3.3 Modified Debye-like Model (Ulaby et al., 2014)

Ulaby et al. (2014) used the same formulation (Eqs. 134-124) from Hallikainen et al., (1986), except they scaled the A parameter in Eq. 134.1 (i.e., the real part of the  $\varepsilon_{eff}$ ) with the  $A_1$  factor (<1) from Eq. 124.1, as follows. The imaginary part,  $\varepsilon''_{eff}$ , of Ulaby et al. (2014) and Hallikainen et al. (1986) remained the same. Here, this model is referred to as the 'Ulaby model'.


205

$$A = A_1(1 + 1.83 \,\rho_{ds} + 0.02 \,A_1 \,v_{w*}^{1.015}) + B_1 \tag{153}$$

The imaginary part,  $\varepsilon''_{aff}$ , of Ulaby et al. (2014) and Hallikainen et al. (1986) remained the same. Here, this model is referred to as the 'Ulaby model'.

## 235 **2.3.4 MEMLS Version 3 (MEMLS V3)**

MEMLS V3 uses the MG rule to compute the effective dielectric constant of wet snow as a two-phase mixture of liquid water inclusions in a dry snow background (host). It uses experimentally determined depolarization factors  $N_a = 0.005$ ,  $N_b = N_c = 0.4975$  (a prolate spheroidal shape of the inclusion) from Hallikainen et al. (1986) and Matzler et al. (1984)

$$\varepsilon_{eff} = \frac{(1-v)\varepsilon_{dS} + v\varepsilon_{w}K}{(1-f) + vK} \tag{14.1}$$

$$K = \frac{1}{2} \left( K_a + K_b + K_c \right) \tag{14.2}$$

$$K_{u} = \frac{\varepsilon_{ds}}{\varepsilon_{ds} + N_{u}(\varepsilon_{uv} - \varepsilon_{ds})} \quad u = a, b, c \tag{14.3}$$

where  $\varepsilon_{w}$  is the dielectric constant of pure water. Here, this model is referred to as MEMLS3.

## 2.3.5 Tinga Model

The Tinga model (Tinga et al., 1973) considers wet snow as a three-phase mixture (air-snow-liquid water) where air is the background and water is a spherical shell surrounding another confocal shell (ice). The effective dielectric constant is then determined following Eq. 7.

#### 2.3.6 Colbeck Model



Based on observations, Colbeck (1980) revised the PVS mixing theory to derive the dielectric constant of wet snow for three distinct cases. The salient feature of this model is that it permits air, ice, and liquid water to form the continuum environment depending on their volume fraction. When both the density and liquid inclusion are low ( $\rho_{ds}$ < 550 kg m<sup>-3</sup> and  $v_w$ LWC < 7.% percent), air is the continuous environment throughout the medium. This regime (Colbeck (1980) case I) called the 'pendular regime' where ice grains form clusters and isolated liquid water resides in the fillets and veins of the grain contacts, describes well the liquid water inclusion in the percolation zone of GrIS. The shape of the fillets (thin and longer) and veins (shorter) is represented by their aspect ratio ( $n = \frac{c}{a} = \frac{c}{b}$ ), which can lie between 1 (spherical) and 10 (needle shaped). However, comparing with the measurements, Colbeck (1980) suggested an average value of n = 3.5 for this case ( $v_w$ LWC < 7.% percent).

As the liquid water inclusions increase beyond 7 percent and there is enough pore space ( $\rho_{ds}$ < 550 kg m<sup>-3</sup> and thus porosity,  $\phi > 0.4$ ), grain clusters break down, and a transition from the pendular regime to another regime called the 'funicular regime' occurs when liquid water becomes continuous throughout the pore space containing isolated air bubbles and rounded ice grains. Although other studies (Denoth, 1982, 1989, 1994; Denoth et al., 1984) report that this transition can occur at lower  $v_w$ LWC (< 7\_\(\frac{94}{2}\) percent), we do not consider this case (Colbeck (1980) case II) as it is not representative of typical melt conditions in the percolation zone. It may represent saturated snow or slush in the GrIS ablation zone.

However, if the density is high ( $\rho_{ds} > 550 \text{ kg m}^{-3}$ , high ice fraction and thus low porosity,  $\phi < 0.4$ ), ice forms the continuum medium and air becomes spherical isolated bubbles, while liquid water still resides in the fillets and veins of the grain clusters for low liquid inclusions ( $v_w$ LWC < 7\_% percent). This case (Colbeck, 1980; case III) is also relevant to percolation zone firn, especially at depth below the seasonal snow. Therefore, we implement Colbeck (1980) cases I (pendular regime) and III (low porosity) using 3 component PVS mixing theory in the following form.

$$\varepsilon_{eff} = \begin{cases} & \text{Three-phase PVS mixing with air background} & \rho_{ds} \leq 550 \text{ kg/}m^3 \\ & \text{Three-phase PVS mixing with ice background} & \rho_{ds} > 550 \text{ kg/}m^3 \end{cases}$$

$$(165.1)$$

For n = 3.5, we used m = 0.072 following Picard et al. (2022), where m is the ratio of the depolarization factors.

$$m = \frac{N_c}{N_a} = \frac{N_c}{N_b}$$
 (156.21)

# 2.3.7 Tiuri Model






Tiuri et al. (1984) used experimental data to measure the complex dielectric constant of wet snow for frequencies 859 MHz

12.6 GHz. Their results suggest that the complex dielectric constant of snow is largely unaffected by the snow structure. In dry snow, the dielectric constant is primarily determined by the density. For wet snow, both the imaginary part  $(\epsilon_s^{\mu})$  and the increase in the real part due to liquid water  $(\epsilon_s^{\nu})$  show a similar dependence on volumetric wetness, which were empirically modeled as:

$$\epsilon_s^t = 1 + 1.7 \,\rho_a + 0.7 \,\rho_a^2 + 8.7 \,W + 70 \,W^2 \tag{16.1}$$

$$\epsilon_{s}^{"} = \frac{f}{10^{2}} (0.9 W + 7.5 W^{2}), \ f = 500 - 1000 MHz$$
 (16.2)

## Among the power-law based models, we considered following three well known models. 2.3.8 Birchak Model

The Birchak et al. (1974) model follows a form of a widely usesd power law relation (Eq. 8) with an exponent,  $\beta = \frac{1}{2}$  in the power-law relation of Eq. 8, and hence this model is also known as 'refractive mixing model'. Sihvola et al. (1985) used a similar model based on Cummings (1952) results and obtained a best fit with  $\beta = 0.4$ . The Looyenga model (Looyenga, 1965)

also follows the power-law relations of the form in Eq. 8. Specifically, it uses  $\beta = \frac{1}{3}$ . We will refer these models as Birchak, Sihvola, and Looyenga models, respectively.

$$\varepsilon_{eff}^{\beta} = (1 - f)\varepsilon_{e}^{\frac{1}{2}} + f\varepsilon_{t}^{\frac{1}{2}} \tag{17}$$

To determine the effective dielectric constant of wet snow, Eq. <u>178</u> can be extended for a three-component mixing with respective volume fraction, but we used <u>Mätzler (2006) model PVS mixing of air and ice for the dielectric mixing of for dry snow, then used Eq. <u>178</u> for water inclusion in a dry snow environment.</u>

# Aside from above models, we considered purely empirical 2.3.9 Sihvola Model

Sihvola et al. (1985) used an exponential model based on (Cummings, 1952) results and obtained a best fit with  $\beta = 0.4$  (in Eq. 8). For wet snow as a three-component mixture, we considered water inclusion in a dry snow environment (i.e., two phase mixture), where the dry snow dielectric constant was derived using PVS mixing of air and ice.

$$\varepsilon_{eff}^{\beta} = (1 - f)\varepsilon_e^{0.4} + f\varepsilon_l^{0.4} \tag{18}$$

# 2.3.10 Looyenga Model



The Looyenga model (Looyenga, 1965) also follows the power law relations of the form in Eq. 8. Specifically, it uses  $\beta = \frac{1}{2}$ ,

$$\varepsilon_{eff}^{\frac{1}{2}} = (1 - f)\varepsilon_{e}^{\frac{1}{2}} + f\varepsilon_{t}^{\frac{1}{2}} \tag{19}$$

Similar to the Birchak model, we used PVS mixing rule to determine dry snow dielectric constant, which is used as the host medium in Eq. 19 for liquid water inclusions.

Tiuri et al. (1984) model that used experimental data to measure the complex dielectric constant of wet snow for frequencies 859 MHz – 12.6 GHz. Their results suggest that the complex dielectric constant of snow is largely unaffected by the snow structure. In dry snow, the dielectric constant is primarily determined by the density. For wet snow, both the imaginary part  $(\varepsilon_s'')$  and the increase in the real part due to liquid water  $(\varepsilon_s')$  show a similar dependence on volumetric wetness, which were empirically modelled as,

$$\varepsilon_s' = 1 + 1.7 \,\rho_d + 0.7 \,\rho_d^2 + 8.7 \,v_w + 70 \,v_w^2 \tag{17.1}$$

$$\varepsilon_{s}^{"} = \frac{f}{10^{9}} (0.9 \, v_{w} + 7.5 \, v_{w}^{2}) \cdot f = 500 - 1000 \, MHz$$
 (17.2)

A summary of the above-mentioned wet snow dielectric mixing models is given in Table 1.

Table 1: Salient features of the ten selected dielectric mixing models.

| Models                                   | Mixing rule                  | Host                                                                     | Key parameters                                                                                                                           | References                             |  |
|------------------------------------------|------------------------------|--------------------------------------------------------------------------|------------------------------------------------------------------------------------------------------------------------------------------|----------------------------------------|--|
| Mätzler MEMLS V3                         | Maxwell Garnett (MG)         | Dry snow                                                                 | Depolarization factors $N_a = 0.005, N_b = N_c = 0.4975$                                                                                 | Mätzler and<br>Wiesmann, (2012)        |  |
| Tinga                                    | Tinga-Voss-<br>Blossey (TVB) | Air                                                                      | Multi-phase mixture of randomly dispersed confocal ellipsoids described by the depolarization factors of the inner and outer ellipsoids. | Tinga et al., (1973)                   |  |
| Debye-like                               |                              |                                                                          | Eq. $910 - 11$ , with $A_1 = A_2 = 1$ , and $B_1 = 0$ (Eq. 13)                                                                           | Hallikainen et al.<br>(1986)           |  |
| Hallikainen<br>(Modified Debye-<br>like) | Bruggeman                    | Symmetric                                                                | Eq. $\underline{910}$ $\underline{11}$ , with $A_1$ , $A_2$ , and $B_1$ were determined from empirical fit (Eq. $1\underline{42}$ )      | Hallikainen et al.<br>(1986)           |  |
| Ulaby (Modified<br>Debye-like)           |                              |                                                                          | Same as Hallikainen et al., (1986), with scaled A by Eq. 153                                                                             | Ulaby et al., (2014)                   |  |
| Colbeck                                  | Polder–van Santen<br>(PVS)   | Air, or ice, or<br>liquid water<br>depending on their<br>volume fraction | Pendular regime and low porosity cases defined by Eq. 165, with aspect ratio, n = 3.5                                                    | Colbeck (1980)<br>Picard et al. (2022) |  |
| Birchak                                  | Power-law relation           |                                                                          | $\beta = \frac{1}{2}$                                                                                                                    | Birchak et al. (1974)                  |  |
| Sihvola                                  | (Eq. 8), used with           | Dry snow                                                                 | $\beta = 0.4$                                                                                                                            | Sihvola et al. (1985)                  |  |
| Looyenga                                 | PVS mixing                   |                                                                          | $\beta = \frac{1}{3}$                                                                                                                    | Looyenga, (1965)                       |  |
| Tiuri                                    | nri Empirical fit            |                                                                          | Assumed to be independent of the snow structure. Eq. 176.                                                                                | Tiuri et al. (1984)                    |  |

# 2.4 Theoretical Penetration Depth

An important quantity of interest for liquid water quantification is the depth of penetration (also known as e-folding depth,  $\delta_p$ ), a depth at which the signal power drops to  $\frac{1}{e}$  times (~37\_percent%) of its initial power at a reference location due to absorption and scattering in the snow and firn. The effective depth from which microwave radiometers receive emissions is usually higher depending on the medium properties although the signal strengths progressively diminish (less than 3\_%percent and 5\_%percent of theirits initial value at depths  $5\delta_p$  and  $3\delta_p$ \_-respectively). The actual depth of penetration also depends on

the signal to noise ratio (SNR) as well as the precision of the radiometer instruments. To estimate the total liquid water amount, the radiometer should receive emissions from the full wet layer. The signal power of an electromagnetic wave propagating through snow/firn is determined by the extinction coefficient  $\kappa_e$  of the medium, which is the sum of the volume absorption and scattering coefficient respectively,  $\kappa_e = \kappa_a + \kappa_s$ . Since the snow grains are much smaller than the L-band wavelength,  $\kappa_s \ll \kappa_{as}$ , the absorption coefficient dominates the extinction,  $\kappa_a \approx \kappa_e$ . Therefore, neglecting scattering losses at L-band for a low volume fraction of liquid water ( $\epsilon \epsilon^{"} \ll \epsilon \epsilon'$ ) in snow/firn, the penetration depth can be approximated following Elachi and Zyl (2021) and Ulaby et al. (2014) as:

$$\delta_p = \frac{1}{\kappa_a} \tag{1820}$$

where  $\kappa_a$  is the wet snow power absorption coefficient given by  $\kappa_a = 2\alpha$  and  $\alpha$  is the attenuation coefficient (Np m<sup>-1</sup>) defined by,

$$\alpha = -k_0 \cdot Im(\sqrt{\varepsilon_{eff}}) \tag{2190}$$

where  $k_0$  is the wave number in vacuum,  $k_0 = \frac{2\pi f}{c}$ , c is the speed of light, and f is the frequency in Hz. Therefore, for a given frequency,  $\delta$  is determined by the effective dielectric constant, depending on the average volume fraction of liquid water content and the density of snow/firn. For L-band, the penetration depth in dry snow is significantly higher (> 100m) (Matzler et al., 1984) depending on the density. However, in wet snow, the liquid water inclusion significantly increases absorption ( $\varepsilon \varepsilon$ ) thus decreasing  $\delta$ .

# 335 2.5 Liquid Water Amount

For a volume fraction of liquid water  $v_w$  (percent%) with a wet layer thickness of  $t_{wet}$ , (m), the LWA is calculated by the product of the two,

$$LWA = v_w t_{wet} \text{ m. w. e}$$
 (240)

We chose to express the *LWA* in [mm], which is equivalent to [kgm<sup>-2</sup>] (because the density of water is 1000 kg m<sup>-3</sup>). The *LWA*represents the vertically integrated liquid water content within the SMAP grid point at that timestamp, corresponding to the SMAP effective sensing depth.

#### 2.6 Liquid Water Retrieval Algorithm

For the LWA retrieval, we iteratively used an inversion-based framework, first minimizing a cost function between the simulated and mean observed TB measured at vertical (p = V) polarization during the frozen season which we considered to span Jan 1 – Mar 31 and Nov 1 - Dec 31 for the pre- and post-summer seasons, respectively, across for the GrIS percolation

zone. For the observations, we used the SMAP enhanced-resolution data products generated using the radiometer form of the Scatterometer Image Reconstruction (rSIR) algorithm (Long et al., 2019; Brodzik et al., 2021) posted on the EASE-2 3.125 km grid (Brodzik et al., 2012; 2014). The rSIR technique utilizes the measurement response function (MRF) of each sample and combines the overlapping (at close but different acquisition times) MRFs to reconstruct an enhanced-resolution TB image (Early and Long, 2001; Long, 2019; Long et al., 1993; Long and Brodzik, 2016; Long and Daum, 1998). The data product provides the twice daily sampling of GrIS in the form of combined morning and evening passes. The advantage of this rSIR processing is that it improves the overall effective resolution of the measurements of about 30 percent compared to the original data products (Long et al., 2023; Zeiger et al., 2024). The radiometric precision of the SMAP original data is within 0.5 K (Chaubell et al., 2018, 2020; Piepmeier et al., 2017).

Using the average measured density from the top 3 meters of snow as recorded from the PROMICE or GC-Net AWS, the algorithm first optimizes the <u>pre-summer</u> baseline scattering coefficient using the <u>pre-summer mean</u> frozen season (Jan 1  $\underline{\hspace{0.005cm}}$  Mar 31) TBs. If the post-summer mean frozen-season TB (November 1  $\underline{\hspace{0.005cm}}$  December 31) is lower than the pre-summer mean frozen-season TB—such as due to crust formation caused by refreezing—the post-summer frozen-season TB is used in the optimization to determine a separate reference for the late melt season. The transition between frozen references is identified by the day on which the maximum TB is observed, as refreezing becomes the dominant process thereafter. However, if the post-summer mean frozen-season TB is higher than the pre-summer mean, indicating a warmer background from remnant melt or latent heat, the pre-melt frozen reference is used throughout the year. With these initial conditions, the melting TB in the summer season is a nonlinear function of the wet layer thickness ( $t_{wet}$ ), liquid water content ( $v_w$ ), and melt-related and other snow firn metamorphisms. Here, we dide not account for melt induced snow metamorphism in the forward simulation, except for the adjustments in the reference TBs. We then used the melt season observed TBs to derive an average wet layer thickness ( $t_{wet}$ ) and liquid water content ( $v_w$ ) in a two-step optimization process. To remove any spurious melt during frozen season, we also derived a threshold-based binary melt flag. The threshold was determined by an algorithm following (Torinesi et al., (2003):

$$Th = TB_{ref} + m * \sigma \tag{21}$$

where  $TB_{ref}$  is frozen reference,  $\sigma$  is the standard deviation of the TB during the pre-summer reference period, and m is an empirically derived constant. For SMAP V-pol TBs over the Greenland percolation zone, we found m = 10 to be optimal.





Figure 1: Configuration of a simplified three-layer ice sheet model to represent equivalent snow and firn stratigraphy for forward modeling of the brightness temperature.

The near-surface density profile in the percolation zone is highly variable and characterized by multi-scale fluctuations (Rennermalm et al., 2022; (Johnson et al., 2014) and the references therein). Seasonal melting and refreezing further complicate this structure, leading to the formation of features such as random ice layers and ice pipes. Accurately modeling these effects across the percolation zone remains a significant challenge due to the lack of detailed ancillary data (e.g., temperature, density) and is an ongoing area of research. These sub-grid-scale structural variabilities contribute to the substantial scattering of L-band TBs, particularly during the frozen season (Hossan et al., 2024). Even with measured ice core profiles, simulating L-band TBs remains difficult due to these complexities.

We implemented a <u>simplified</u> three-layer ice sheet configuration (Figure 1) to simulate TB based on MEMLS V3 (Mätzler and Wiesmann, 2012). <u>In this setup (Figure 1), t</u>The top layer holds the dry snow/firn during frozen season and liquid water during melt season. The bottom layer of the three-layer configuration is <u>defined as</u> semi-infinite ice. <u>In the upper layers of the percolation zone</u>, the density profile is highly variable with discrete ice layers and ice pipes (Rennermalm et al., 2022), that cause significant scattering (internal reflections) of frozen season microwave emissions. To account for the combined reflective effects by the complex stratigraphy due to ice layers, we designate the middle layer (underneath the dry/wet snow layer) as a highly reflective layer by specifying its dielectric constant with a high real part ( $\varepsilon_{\pm}$ ) that varies spatially. introduced a middle layer modelled as an equivalent dielectric slab with a high, spatially variable real permittivity. This layer is designed to represent the bulk reflectivity caused by complex firn stratigraphy, following a similar approach by Mousavi et al. (2022). It does not contain liquid water and is not intended to reflect any specific physical layer, but rather to simulate the integrated

dielectric contrast beneath the seasonal snowpack. Its properties remain constant over time at each grid point and have negligible influence on liquid water retrievals, which are governed by the top layer's dynamic properties.







In MEMLS each layer is defined by its thickness (d), physical temperature (T), density ( $\rho$ ), volumetric liquid-water content ( $v_w$ ), exponential correlation length ( $l_{ex}$ ), and salinity (S). Since our objective is to evaluate the relative behaviours of dielectric mixing models, we tried to make the model as simple as possible by assuming reasonable ranges and fixed values of possible parameters. The top layer has thickness 0.1–20 m, volumetric liquid-water content of 0–6 percent, and a fixed temperature of 250 K and 273.15 K when  $m_v = 0$  (dry) and  $m_v > 0$  (melt) respectively.

While, L-band TB is sensitive to dry snow density (Hossan et al., 2024; Houtz et al., 2019), we used the average measured density in the top 3 m to better constrain the retrievals. Considering the insensitivity of L-band measurements to snow microstructure, we set  $l_{ex} = 0$  mm for all three layers (Schwank et al., 2014). Planar interfaces and specular reflection are assumed. For ice sheets, the salinity can also be set to 0 ppt. The range of volume fraction of liquid water was determined based on earlier Experiments (Coléou and Lesaffre, 1998; Colbeck, 1974), which suggest that the irreducible water saturation because of capillary retention ranges between 6.5 - 8.5 percent of the pore volume depending on the density. Considering snow/firn density in the percolation zone, we determined the maximum volume fraction is within 6 percent. Exceptions to this are the saturated water such as buried and open lakes (Dunmire et al., 2021), firn aquifers (Miller et al., 2022b,  $\frac{2018}{2020b}$ , which are atypical for this area and we did not include these cases into our consideration

The middle layer has a fixed thickness of 5 m. It shares the same physical temperature as the top layer during the frozen season and uses a fixed temperature of 265 K when the top layer contains liquid water (at melting point). The real part of its relative dielectric constant varies between 5 26 is tuned while the imaginary part is fixed 0.0002 (same as ice loss factor). The bottom layer is semi-infinite ice with fixed density (917 kg m<sup>-3</sup>) and physical temperature of 255 K regardless of dry and melt season. We also considered the cosmic background radiation (Tc = 2.7 K). However, we did not consider any correction for the atmospheric contribution, because it is small compared to the melt signal (approximately at most 2 K at L-band frequencies (Houtz, et al. 2019)). For a fixed location, the same layer characteristics were applied to all the models.

MEMLS utilizes the six-flux theory to model volume scattering and absorption. It also accounts for the effects of radiation trapping due to internal reflection, along with the coherent and incoherent reflections at the layer interfaces (Mätzler and Wiesmann, 2012). The model uses an empirical approach to compute the scattering coefficient, while the absorption coefficient, refraction, and reflection at layer boundaries are derived using physical models. For effective permittivity, by default, the <u>updated\_latest\_version\_(V3)</u> of MEMLS considers wet snow as a two-phase mixture of prolate ellipsoidal liquid water inclusions in a dry snow background and uses the MG dielectric mixing rule with depolarization factors from Hallikainen et al. (19864) as mentioned in the previous section. We refer to this default configuration of MEMLS V3 as simply <u>Mätzler model\_MEMLS3</u>. For other models, we used the same setup and input parameters, except we changed the formulations for the complex wet snow dielectric constants (Sec. 2.3).

# 2.7 LWA Estimates from a Surface Energy and Mass Balance Model

As references, we considered independent LWA estimates from two ice sheet SEMB models, namely the Samimi et al., (2021) SEMB model, which was locally calibrated (Samimi et al., 2020, 2021; Ebrahimi and Marshall, 2016;) and the Glacier Energy and Mass Balance (GEMB) model (Gardner et al., 2023), within the NASA Ice-sheet and Sea-Level System Model (ISSM). Both models were forced with the in situ automatic weather stations (AWS) measurements from the Programme for Monitoring of the Greenland Ice Sheet (PROMICE) and Greenland Climate Network (GC-Net) located in the percolation zone of the GrIS. These models used averaged hourly observations of air temperature, air pressure, upwelling and downwelling short and longwave radiation fluxes, snow-surface height, wind speeds (Fausto et al., 2021) along with subsurface profiles of temperature, density, and stratigraphy for initializations (Vandecrux et al., 2023). The SEMB models determine the net energy available for melting if the surface temperature is at the melting point, otherwise for warming or cooling the snow in the upper layer. The subsurface temperature and density then evolve within a one-dimensional model, which is coupled with hydrological processes like meltwater infiltration, refreezing, and retention within the firm. Although the two models under consideration used the same forcing, they use separate parameterizations for these physical processes and a separate model configuration. We refer the reader to Samimi et al., (2021), and Gardner et al., (2023), for specific model details. Despite limitations, these SEMB models are currently the most viable way of validating satellite retrievals. For comparison, we consider these two models individually as well as their ensemble (average).

## 3 Results







## 3.1 Effective Complex Dielectric Constant

Liquid water increases both the real and imaginary part of the dielectric constant of snow/firn. Since the dry snow has negligible loss factors at microwave frequencies, almost all the changes in imaginary parts come from liquid water inclusion. Figure 2 shows the change of the complex dielectric constant at a fixed density of 400 kg m<sup>-3</sup> as function of the volume fraction  $(v_w)$  of liquid water inclusions for up to 6 percent as appropriate for the percolation zone.

There are large spreads between the models for both the real (Fig. 2a and 2b) and imaginary parts (Fig 2c and 2d); this spread increases as  $v_w$ LWC increases. The deviations between the models are higher for the imaginary part than in the real part. For  $v_w$ LWC < 2 percent, the models' agreements for the real part (Fig. 2ab) are consistent within two tenths of the relative dielectric constant, except the Hallikainen model (dark greendashed black line), which appeared an outlier. The Ulaby model (red solid blue line) uses a scaling factor of A<1, resulting in the lowest  $v_w$ LWC estimate among the models up to about 2 percent (Fig. 2ba); at higher  $v_w$ LWC values it provides an intermediate estimate (Fig. 2ab). The Ulaby, Tinga, and Debyelike models provide real part of dielectric constant lower than that of even dry snow (the dashed grey line indicates the permittivity of dry snow at -0.5 C with the Mätzler, 2006 model) for up to  $v_w$ LWC 1.2, 0.7, and 0.4 percent, respectively. The Debye-like model and the low frequency approximate of the Hallikainen model closely agree with the empirical Tiuri model,

and both lie in the upper end for the  $v_w$ LWC > 2 percent range, while the <u>Mätzler model (solid black line)</u>MEMLS3 predicts an intermediate result for the entire  $v_w$ LWC range. The exponential models (Looyenga, Sihvola, and Birchak) reasonably agree with the median model (e.g., <u>Mätzler MEMLS3</u>) for low  $v_w$ LWC (< 2 <u>percent</u>%) and stay in the lower end of the curves for the higher  $v_w$ LWC. Among the structure dependent models, the Colbeck and Tinga models provide relatively lower estimates of the real part of the dielectric constant and agree with the exponential models, especially for  $v_w$ LWC > 2 percent.




Figure 2: Change of real (top panel) and imaginary (bottom panel) parts of complex dielectric constant of snow/firn of a fixed density of 400 kg m<sup>-3</sup> as function of volume fraction ( $v_w$ ) of liquid water content. The <u>rightleft</u> panel shows a zoomed version of the right panel for LWA range 0 - 2 percent. As reference, real (top) and imaginary (bottom) parts of complex dielectric constant of dry snow at the same density and snow temperature of -0.5 C are shown by <u>horizontal</u> grey dashed lines.

For the imaginary part of the snow/firn dielectric constant, the Debye-like, Hallikainen, and Ulaby models, which are the same group of models with modifications for frequency dependencies, generally follow the empirically derived Tiuri model almost for the entire range of  $v_w$ LWC under consideration. However, it is worth noting that small differences in the loss factor, especially in the lower end, can result in significant differences in terms of TB and depth of penetration. The Hallikainen and

Ulaby models are the same for the imaginary part, overlapping with each other. The Tinga model provides the highest estimate of the loss factor for  $v_w$ LWC up to about 2 percent, then it falls exponentially for the higher end. Mätzler modelMEMLS3 again provides an intermediate estimate of the loss factor for the entire  $v_w$ LWC range under consideration. The exponential models stay in the lower end, as in the case of the real part, with the lower value of  $\beta$ , giving the lower estimate. The Colbeck model results consistently in the lowest value of the imaginary part of the dielectric constant for the entire range of  $v_w$ LWC. Since the density is less than 550 kg m<sup>-3</sup> for these curves, it included only Case I (pendular regime) of Colbeck (1980).

Figure 3: Penetration depth at L-band (1.41 GHz) for a snow/firn density of 400 kg m<sup>-3</sup> as function of volume fraction ( $v_w$ ) of liquid water content (a). (b) a zoomed version of (a) for LWA range 3 – 6 percent %.

# 3.2 Penetration Depth


The differences in the imaginary part of the dielectric constants are manifested in the penetration depth, an important variable for liquid water quantification. Figure 3 illustrates the effective penetration depth of L-band (1.41 GHz) signals

estimated by the models for a snow/firn density of 400 kg m<sup>-3</sup> as a function of the  $v_w$ LWC in the 0\_-6 percent range (Fig. 3a) and the 3\_-6 percent range (Fig. 3b). All models predict an exponential decay of effective penetration, but they exhibit substantial differences with respect to one another, though the range generally reduces with increasing  $v_w$ LWC. For  $v_w$ LWC of 1, 3, and 5% percent, the model estimates of penetration depth range between 2.8\_-12.8 m, 1\_-4 m, 0.5\_-2.3 m respectively. The Tinga and Ulaby models provide the lowest estimate of penetration depth for  $v_w$ LWC < 2 percent, and  $v_w$ LWC > 2 percent ranges respectively, while the Colbeck model gives the highest estimate for the entire  $v_w$ LWC range as it estimates the lowest loss factor among all the models. The Debye-like, Hallikainen, and Ulaby models closely follow each other. Mätzler modelMEMLS3 provides an intermediate estimate of penetration depth. The empirical Tiuri model aligns with Mätzler modelMEMLS3 for low  $v_w$ LWC (< 1\_% percent); however, it matches better with Debye-like, Hallikainen, and Ulaby models for higher  $v_w$ LWC (> 2 % percent). The exponential models, consistent with their complex dielectric constant, lie in between.

## 3.3 Simulated Brightness Temperature

TB simulated with these models for a typical representative snowpack in the percolation zone are illustrated as function of  $v_w$ LWC (bottom x-axis) and LWA (top x-axis) in Figure 4 for wet layer thicknesses of 1 m (Fig. 4a, 4d), 2 m (Fig. 4b, 4e), and 3 m (Fig. 4c, 4f), respectively. For the V-pol, the Tinga model appears to be the most sensitive for low  $v_w$ LWC and LWA, then it gradually slows down as the LWA increases, when the Ulaby model provides the highest TB. The Debye-like model closely follows the Ulaby model; the Hallikainen model, which uses the same imaginary part as the Ulaby model, but a higher real part of the dielectric constant, shows lower TB estimates. The difference that is also function of LWA, is more obvious in the H-pol results (Fig. 4d\_-4f). The Tiuri and MätzlerMEMLS3 models produce higher TB projections than the Ulaby and Debye-like models for the lower range of LWA, but it flipped for higher range of LWA, and the transition depends on the thickness of the wet layer. In line with the complex dielectric constant, the Colbeck model provides the lowest estimates for almost the entire LWA range under consideration (except for LWA > 150 mm, H-pol), and the Birchak, Sihvola, and Looyenga models offer moderate values for all cases. Although, the changes are more pronounced in the H-pol TB, the trends with the LWA are similar except that the saturation in TB occurs for relatively lower LWA compared to the V-pol signals, especially for thicker wet layers (Fig. 4e\_-4f).

The results depend on the density of the dry snow (porosity), which are shown for three different densities (200 kg m<sup>-3</sup> (Fig. 5a, 5d), 400 kg m<sup>-3</sup> (Fig. 5b, 5e), and 600 kg m<sup>-3</sup> (Fig. 5c, 5f)) for a fixed thickness of wet layer (2 m) in Figure 5. The Debye-based models (Debye-like, Hallikainen, and Ulaby) along with the Tiuri model show significantly higher sensitivity with  $v_w$  LWC and thus provide lower estimates of LWA than the MätzlerMEMLS3 and Tinga models for the low snow density (200 kg m<sup>-3</sup>) at both V- and H-pol results. However, for high snow density (600 kg m<sup>-3</sup>), this is reversed, Tinga and Mätzler models MEMLS3 exhibit higher sensitivity and provide lower estimates of LWA than the Debye-based and Tiuri models, while at intermediate density (400 kg m<sup>-3</sup>), they agree closer for both V- and H-pol TB. Although the sensitivity of the rest of the models varies with the dry snow density, they consistently demonstrate lower sensitivity and provide higher estimates of LWA than the above-mentioned models across the complete density range.

Figure 6 depicts simulated TB, like Figure 4, but as a function of wet layer thickness in (bottom x-axis) and LWA (top x-axis) at a fixed snow/firn density of 400 kg m<sup>-3</sup> for three cases of fixed  $v_w$  LWC of 1 percent (Fig. 6a and 6d), 2 percent (Fig. 6b and 6e), and 3 percent (Fig. 6c and 6f). In a broader perspective, the trends of TB with the thickness of the wet layer at a fixed  $v_w$  are similar to the TB trends with  $v_w$  at the fixed thickness of the wet layer, as presented in Fig. 4. TB grows exponentially with both  $v_w$  and  $t_{wet}$ , where each model has a different growth factor, which also depends on  $v_w$  and  $t_{wet}$  themselves along with dry snow density and other background conditions.



Figure 4: Vertically (a-c) and horizontally (d-f) polarized brightness temperature at L-band for a snow/firn density of 400 kg m<sup>-3</sup> as function of volume fraction  $(v_w)$  of liquid water content in percent (bottom x-axis) and total liquid water amount in mm (top x-axis) simulated with Mätzler model MEMLS3 using different wet snow mixing models for three wet layer thickness: 1 m (a and d), 2 m (b and e), and 3 m (c and f).

Figure 5: Vertically (a-c) and horizontally (d-f) polarized brightness temperature at L-band for a wet layer thickness of 2<sub>m</sub> as function of volume fraction ( $v_w$ ) of liquid water content in percent % (bottom x-axis) and total liquid water amount in mm (top x-axis) simulated with Mätzler modelMEMLS3 using different wet snow mixing models for three snow/firn densities: 200 kg m<sup>-3</sup> (a and d), 400 kg m<sup>-3</sup> (b and e), and 600 kg m<sup>-3</sup> (c and f).

Figure 6: Vertically (a-c) and horizontally (d-f) polarized brightness temperature at L-band for a snow/firn density of 400 kg m<sup>-3</sup> as function of wet layer thickness (bottom x-axis) and total liquid water amount in mm (top x-axis) simulated with Mätzler modelMEMLS3 using different wet snow mixing models for three fixed volume fraction ( $v_w$ ) of liquid water content: 1 percent (a and d), 2 percent (b and e), and 3 percent (c and f).

# 3.4 Brightness Temperature Sensitivity to Liquid Water Change



The sensitivity of TB to LWA change decreases with increasing LWA. We compute the change of TB for every 1 mm change in LWA, which is shown for V- and H-pol TB in Figure 7. Here we considered a  $v_w$ LWC of 3 percent, and increased the wet layer thickness from 0.1 m to 5 m. The sensitivities at H-pol are higher than at V-pol for all models. The sensitivity of TB to the change of LWA decays exponentially across all models, falling below 1 K mm<sup>-1</sup> at < 50 mm of LWA for V- and H-pol. For models that demonstrate higher sensitivity for lower LWA, the sensitivity declines more sharply and saturates at relatively lower LWA (after which they show negative sensitivity, i.e., TB decreases as LWA increases; however, we did not consider negative sensitivity regime in this manuscript as this happens at oversaturated LWA LWC amounts-not typical for the percolation zone of the Greenland Ice Sheet, see Section 1).

The Ulaby and Tiuri models show the highest sensitivity at lower LWA, closely followed by the Hallikainen, Tinga, and <u>MätzlerMEMLS3</u> models. The sensitivities of these models fall below tenths of K mm<sup>-1</sup> for LWA > 70 mm for both V-

and H-pol. The Birchak, Sihvola, and Looyenga models demonstrate moderate sensitivities, while the Colbeck model presents the lowest sensitivity among the models for lower LWA; however, their sensitivities also decrease slowly with LWA, and they remain reasonably sensitive for the higher end of the LWA. Although, the first group of models shows almost negligible sensitivities close to or beyond 100 mm of LWA, no models showed perfect 0 or negative sensitivities within 150 mm of LWA. However, it is obvious that for a majority of the models the uncertainty of the retrievals at LWA > 60\_-\_70 mm will be significantly higher.

Figure 7: L-band vertically (a) and horizontally (b) polarized brightness temperature sensitivity (change of TB in K per mm change in liquid water amount) at a snow/firn density of 400 kg m<sup>-3</sup> as of total liquid water amount in mm simulated with Mätzler model MEMLS3 using different wet snow mixing models for a fixed 3 percent volume fraction  $(v_w)$  of liquid water content with varying wet layer thickness. Top panel shows the results for a LWA range 0-50 mm, and the bottom panel shows the results for an extended range (0-100 mm).

## 3.5 LWA Retrievals


Figure 8 presents compares the simulated V-pol TBs with the observed V-pol SMAP TB time series (blue dash-dottedack solid line) compared to along with the mean frozen season TB references (dotted blue dashed grey lines), during the 2023 melt season at six PROMICE and GC-Net AWS. The locations AWS were selected based on their varying geographic

locations in the percolation zone, climatic record of melt, and in situ data availability for the validation. -Table 2 shows the geographic locations, elevations along with the mean 2 m air temperature, the mean dry snow density of the upper 3-\_m, and the mean pre-summer frozen season V-pol L-band TB at these sites.

Figure 8: L-band vertically polarized <u>observed and simulated</u> brightness temperature <u>time series observations</u> during 2023 melt season at six selected PROMICE and GC-Net within the GrIS automatic weather stations (AWS) locations within the percolation areazone. Frozen season TB references are shown as dashed grey lines.



The 2023 melt year experienced above average melting (Poinar et al., 2023), though the annual average temperatures reflect both a cooler accumulation period and warmer melt season (Poinar et al., 2023). All sites have very stable but different frozen season TB, representative of their different subsurface backgrounds. Some occasionally decreasing spikes before and after the melt season at SDM site are noise because of rSIR processing (Long and Brodzik 2024, confirmed through personal communications). Since the simulated TBs were optimized with the mean observed TBs during frozen season, they are less affected by these noise during frozen season. Throughout the summer season at each of the AWS location, the simulations with different models closely align with the observations with no significant bias. The agreements are so close that the simulated TBs are almost overlaid by the observed TBs (Fig. 8). But this was obtained with different combinations of  $t_{wet}$  and  $v_{w}$  resulting different LWA.

CP1 and DY2 are both perfect representatives of the Greenland percolation zone, with moderate upper layer density and numerous ice layers and pipes due to annual refreezing of seasonal melt (Jezek et al., 2018; Vandecrux et al., 2023). These

ice layers significantly attenuate microwave emissions from deeper layers giving very low frozen season TBs (~148 and 144 K<sub>2</sub> respectively). At the same time, these sites also provide high TB sensitivity to liquid water during peak melt season as the effective emissions approaches close to unity (>0.95, see Fig. 8<u>a-b</u>). <u>At both AWS locations, the TBs remained elevated through October compared to their pre-summer frozen references.</u>

**Table 2.** Summary of the six selected PROMICE and GC-Net automatic weather station (AWS) locations. The location information was adopted from PROMICE and GC-Net Automatic Weather Station metadata, available at <a href="https://promice.dk">https://promice.dk</a>, last accessed on 11/27/2024. Mean annual, summer (June – August) air temperature, mean dry snow density, and mean frozen season brightness temperature were calculated using 2023 data.

| SitesSite ID | Latitude<br>(degrees<br>north) | Longitude<br>(degrees<br>east) | The altitude<br>above mean<br>sea level<br>(m) | Annual Mean of 2 m air temperature (°C) | Summer<br>(JJA)<br>Mean of 2<br>m air<br>temperature<br>(°C) | Mean dry<br>snow<br>density of<br>upper 3 m<br>(kg m <sup>-3</sup> ) | Mean<br>frozen<br>season V-<br>pol TB (K) | Maximum<br>summer V-<br>pol TB (K) |
|--------------|--------------------------------|--------------------------------|------------------------------------------------|-----------------------------------------|--------------------------------------------------------------|----------------------------------------------------------------------|-------------------------------------------|------------------------------------|
| CP1          | 69.87                          | -47.04                         | 1950                                           | -17.41                                  | -4.84                                                        | 440                                                                  | 148.5                                     | 259                                |
| DY2          | 66.48                          | -46.3                          | 2127                                           | -16.8                                   | -4.35                                                        | 460                                                                  | 144.5                                     | 258.5                              |
| KAN_U        | 67                             | -47.04                         | 1848                                           | -10.79                                  | -2.20                                                        | 480                                                                  | 204                                       | 251.7                              |
| NSE          | 66.48                          | -42.5                          | 2387                                           | -16.22                                  | -5.90                                                        | 440                                                                  | 195                                       | 252.7                              |
| SDL          | 66                             | -44.5                          | 2475                                           | -14.73                                  | -5.55                                                        | 420                                                                  | 167                                       | 254.4                              |
| SDM          | 63.15                          | -44.82                         | 2898                                           | -13.79                                  | -6.02                                                        | 420                                                                  | 185                                       | 254.2                              |

**Table 3.** Average thickness of wet layer (in cm) during 2023 melt season (May – Sept) retrieved by ten dielectric mixing models with SMAP observations and two surface energy and mass balance models forced by in situ observations at the six selected PROMICE and GC-Net automatic weather station (AWS) locations.

| Sites | <u>Mätzler</u> | Tinga                     | Debye-like                | Hallikainen               | Ulaby          | Colbeck        | Birchak                   | Sihvola                   | Looyenga                      | <u>Tiuri</u> | SAMIMI | <u>GEMB</u> |
|-------|----------------|---------------------------|---------------------------|---------------------------|----------------|----------------|---------------------------|---------------------------|-------------------------------|--------------|--------|-------------|
| CP1   | <u>117</u>     | <u>8392</u>               | <u>205</u> 155            | <u>141</u> <del>137</del> | <u>205</u> 161 | <u>315</u> 277 | <u>151</u> <del>180</del> | <u>160</u> 228            | <u>170223</u>                 | <u>122</u>   | 193    | <u>179</u>  |
| DY2   | <u>177</u>     | <u>100</u> <del>113</del> | <u>173</u> <del>161</del> | <u>177226</u>             | <u>239</u> 194 | <u>281</u> 317 | <u>131</u> <del>163</del> | <u>201</u> 221            | <u>272<del>236</del></u>      | <u>194</u>   | 177    | <u>169</u>  |
| KAN_U | <u>149</u>     | <u>62</u> 90              | <u>152</u> <del>188</del> | <u>175</u> 200            | <u>134</u> 212 | <u>180</u> 226 | <u>165</u> 145            | <u>187</u> <del>165</del> | <u>202</u> 257                | <u>139</u>   | 99     | <u>79</u>   |
| NSE   | <u>114</u>     | <u>88</u> 91              | <u>208</u> 98             | <u>288</u> 193            | <u>158</u> 244 | <u>268</u> 231 | <u>184208</u>             | <u>181</u> <del>163</del> | <u>231</u> 200                | 104          | 77     | <u>100</u>  |
| SDL   | <u>116</u>     | <u>143</u> 136            | <u>153</u> 158            | <u>258</u> 252            | <u>165</u> 142 | <u>345</u> 299 | <u>212</u> <del>195</del> | <u>169250</u>             | <del>273</del> <del>331</del> | <u>172</u>   | 185    | <u>122</u>  |
| SDM   | <u>117</u>     | <u>87</u> <del>108</del>  | <u>156</u> 174            | <u>164</u> 154            | <u>225</u> 158 | <u>294276</u>  | <u>140156</u>             | <u>165222</u>             | <u>214</u> 189                | <u>154</u>   | 119    | <u>79</u>   |

**Table 4.** Maximum summer melt (in mm) during 2023 melt season (May – Sept) estimated by ten dielectric mixing models with SMAP observations and two surface energy and mass balance models forced by in situ observations at the six selected PROMICE and GC-Net automatic weather station (AWS) locations.

| Sites | Mätzler   | Tinga                   | Debye-like   | Hallikainen             | Ulaby        | Colbeck                   | Birchak      | Sihvola                 | Looyenga                  | <u>Tiuri</u> | SAMIMI | GEMB |
|-------|-----------|-------------------------|--------------|-------------------------|--------------|---------------------------|--------------|-------------------------|---------------------------|--------------|--------|------|
| CP1   | 44        | <u>50</u> 55            | <u>39</u> 38 | <u>41</u> 42            | <u>39</u> 38 | <u>114</u> 116            | <u>60</u> 60 | <u>86</u> 86            | <u>102</u> <del>110</del> | <u>37</u>    | 76     | 97   |
| DY2   | <u>37</u> | <u>39</u> 36            | <u>37</u> 37 | <u>41</u> 43            | <u>38</u> 37 | <u>101</u> <del>104</del> | <u>54</u> 55 | <u>76</u> 77            | <u>96</u> 99              | <u>36</u>    | 64     | 91   |
| KAN_U | <u>21</u> | <u>21</u> <del>17</del> | <u>24</u> 26 | <u>28</u> 29            | <u>24</u> 26 | <u>60</u> 62              | <u>31</u> 32 | <u>45</u> 46            | <u>57</u> 58              | <u>23</u>    | 30     | 45   |
| NSE   | <u>25</u> | <u>22</u> 23            | <u>27</u> 24 | <u>32</u> <del>30</del> | <u>25</u> 29 | <u>73</u> 75              | <u>35</u> 37 | <u>52</u> 54            | <u>66</u> 70              | <u>23</u>    | 32     | 70   |
| SDL   | <u>33</u> | <u>25</u> 26            | <u>30</u> 30 | <u>36</u> 36            | <u>30</u> 29 | <u>95</u> 95              | <u>45</u> 46 | <u>67</u> <del>67</del> | <u>84</u> 85              | <u>28</u>    | 77     | 69   |
| SDM   | <u>31</u> | <u>33</u> 27            | <u>29</u> 29 | <u>32</u> 32            | <u>30</u> 29 | <u>90</u> 90              | <u>44</u> 44 | <u>6463</u>             | <u>82</u> 82              | <u>27</u>    | 52     | 55   |



KAN\_U is located at the lowest elevation (1848 m) of all the sites close to the equilibrium line in the southwestern Greenland. Air temperatures are often above freezing during the melt season (Table 2), and the region experiences substantial surface melting. As a result, ice layers are thicker, and the near surface densities are high with low variability. Frozen season TB is the highest of all sites examined, and during 2023 melt season (Table 2), TB is possibly saturated due to extensive persistent melt that keeps TB elevated beyond the end of SeptemberOctober (Fig. 8c).

NSE, SDL, and SDM are located at high elevation in southeast Greenland. These locations generally receive more accumulation and less melt than the other stations examined here (Fausto et al., 2021). Upper layer densities are low to moderate with lesser number of ice layers. This is revealed by their moderate means frozen season TBs (Table 2). Contemporaneous large summer peaks of TB at these three sites in 2023 are indicative of melt events (Hossan et al., 2024). However, the duration of the melt events at these AWS locations is substantially shorter than at the previous three AWS locations. In addition, the post-summer mean frozen-season TBs drop below their pre-summer frozen references (note the changes in the reference TBs, Fig. 8d-f), unlike at the previous three sites. This is because the post-melt temperature at these locations drops sharply, possibly due to crust formation that enhances internal reflections.

Table 5. Pearson linear correlation coefficient between LWA estimate by each of the dielectric mixing model and their ensemble with SMAP observations during 2023 melt season (May – Sept) and corresponding LWA estimate obtained by averaging Samimi and GEMB surface energy and mass balance models forced by in situ observations at the six selected PROMICE and GC-Net automatic weather station (AWS) locations.

| Sites | <u>Mätzler</u> | Tinga                       | Debye-<br>like   | Hallikainen      | Ulaby                       | Colbeck          | Birchak                     | Sihvola                     | Looyenga                    | <u>Tiuri</u> | Ensemble         |
|-------|----------------|-----------------------------|------------------|------------------|-----------------------------|------------------|-----------------------------|-----------------------------|-----------------------------|--------------|------------------|
| CP1   | 0.79           | <u>0.74</u> 0.74            | 0.840.84         | 0.830.83         | 0.850.84                    | 0.800.80         | <u>0.79</u> 0.79            | 0.800.79                    | <u>0.80</u> 0.80            | 0.81         | <u>0.80</u> 0.80 |
| DY2   | 0.97           | <u>0.95</u> 0.96            | 0.980.98         | <u>0.98</u> 0.98 | 0.980.98                    | 0.980.98         | <u>0.97</u> 0.97            | <u>0.97</u> 0.98            | <u>0.97</u> 0.98            | 0.98         | <u>0.98</u> 0.98 |
| KAN_U | 0.70           | <u>0.71</u> <del>0.71</del> | <u>0.67</u> 0.68 | <u>0.68</u> 0.68 | <u>0.67</u> <del>0.67</del> | <u>0.70</u> 0.70 | <u>0.70</u> <del>0.71</del> | <u>0.70</u> <del>0.71</del> | <u>0.70</u> <del>0.71</del> | 0.70         | <u>0.70</u> 0.70 |

| NSE     | 0.92 | 0.900.88         | <u>0.93</u> <del>0.91</del> | <u>0.93</u> 0.91 | <u>0.93</u> 0.91 | <u>0.92</u> 0.90 | <u>0.92</u> 0.90            | <u>0.92</u> <del>0.90</del> | <u>0.92</u> 0.90            | 0.92 | <u>0.92</u> 0.90 |
|---------|------|------------------|-----------------------------|------------------|------------------|------------------|-----------------------------|-----------------------------|-----------------------------|------|------------------|
| SDL     | 0.83 | 0.830.82         | <u>0.85</u> 0.84            | <u>0.85</u> 0.84 | <u>0.85</u> 0.84 | 0.840.83         | 0.840.83                    | 0.840.83                    | <u>0.84</u> 0.83            | 0.84 | <u>0.84</u> 0.83 |
| SDM     | 0.92 | <u>0.90</u> 0.90 | <u>0.93</u> <del>0.93</del> | <u>0.93</u> 0.93 | <u>0.94</u> 0.93 | <u>0.92</u> 0.92 | <u>0.92</u> <del>0.91</del> | <u>0.92</u> <del>0.91</del> | <u>0.92</u> <del>0.91</del> | 0.92 | <u>0.92</u> 0.92 |
| Overall | 0.86 | <u>0.84</u> 0.84 | <u>0.87</u> <del>0.86</del> | <u>0.87</u> 0.86 | <u>0.87</u> 0.86 | <u>0.86</u> 0.86 | <u>0.86</u> 0.85            | 0.860.85                    | <u>0.86</u> 0.85            | 0.86 | <u>0.86</u> 0.86 |

LWA retrieved from L-band TB using different dielectric mixing models and two SEMB models for 2023 melt season at <u>the</u> selected AWS locations are presented in Figure 9. For all the mixing models, the retrieved average thicknesses of wet layer used in the LWA retrieval are given in Table 3.





Regarding the onset of melt season at CP1, the satellite retrievals with different mixing models are the same (Jun 24<sup>th</sup>) and reasonably contemporary with the SEMB models considering noise levels of in in situ instruments. We used the SMAP melt flags (see Sec. 2) to remove spurious melts in winter. Similar flagging for SEMB models is difficult to find. Some studies have used thresholds in LWA (e.g., > 2 mm in Hossan et al., 2024; Zhang et al., 2023; Leduc-Leballeur et al., 2020; Van Den Broeke et al., 2010). However, we did not use such threshold because dielectric mixing models that show higher sensitivity for light melt events can result LWA within 2 mm. Hence, here we rather focus on relative intensities and durations of significant and consistent melt events qualitatively, ignoring suspicious melt events, like one in early June at CP1 where GEMB estimated a short-lived melt (~2.5 mm in amount) but the SAMIMI model like all the satellite retrievals indicates no melt even when both models used the same in situ measurements.

**Table 6.** Mean root mean squared differences (RMSD in mm) between LWA estimate by each of the dielectric mixing model and their ensemble with SMAP observations during 2023 melt season (May – Sept) and corresponding LWA estimate obtained by averaging Samimi and GEMB surface energy and mass balance models forced by in situ observations at the six selected PROMICE and GC-Net automatic weather station (AWS) locations.

| Sites   | <u>Mätzler</u> | Tinga                   | Debye-like             | Hallikainen             | Ulaby                  | Colbeck        | Birchak                    | Sihvola                    | Looyenga                   | <u>Tiuri</u> | Ensemble        |
|---------|----------------|-------------------------|------------------------|-------------------------|------------------------|----------------|----------------------------|----------------------------|----------------------------|--------------|-----------------|
| CP1     | <u>23</u>      | <u>23</u> 24            | <u>22<del>23</del></u> | <u>22<del>22</del></u>  | <u>22<del>23</del></u> | <u>1919</u>    | <u>20</u> 20               | <u>18</u> 18               | <u>18</u> 18               | <u>24</u>    | <u>20</u> 20    |
| DY2     | <u>15</u>      | <u>17</u> <del>17</del> | <u>1413</u>            | <u>12</u> <del>11</del> | <u>13</u> 13           | <u>1415</u>    | <u>9</u> 9                 | <u>5</u> €                 | <u>11</u> <del>12</del>    | <u>15</u>    | <u>8</u> 7      |
| KAN_U   | <u>8</u>       | <u>9</u> 9              | <u>8</u> 8             | <u>9</u> 9              | <u>8</u> 9             | <u>20</u> 20   | <u>8</u> 8                 | <u>13</u> 13               | <u>18</u> 18               | <u>8</u>     | <u>9</u> 9      |
| NSE     | <u>10</u>      | <u>11</u> 44            | <u>910</u>             | <u>8</u> 8              | <u>9</u> 9             | <u>8</u> 8     | <u>8</u> 8                 | <u>6</u> 6                 | <u>7</u> 7                 | <u>10</u>    | <u>7</u> 7      |
| SDL     | <u>21</u>      | <u>23</u> 23            | <u>21</u> 21           | <u>1920</u>             | <u>21<del>21</del></u> | <u>16</u> 16   | <u>18</u> 18               | <u>15</u> 16               | <u>15</u> 16               | <u>22</u>    | <u>18</u> 18    |
| SDM     | <u>10</u>      | <u>11</u> 44            | <u>9</u> 9             | <u>9</u> 9              | <u>9</u> 9             | <u>10</u> 11   | <u>7</u> 7                 | <u>6</u> 7                 | <u>8</u> 9                 | <u>10</u>    | <u>7</u> 7      |
| Overall | <u>15</u>      | <u>16</u> 15.90         | <u>1414.06</u>         | <u>13</u> 13.13         | <u>14</u> 13.88        | <u>1414.98</u> | <u>12</u> <del>11.87</del> | <u>11</u> <del>10.82</del> | <u>13</u> <del>13.38</del> | <u>15</u>    | <u>11</u> 11.48 |

All three methods (SAMIMI, GEMB, and satellite retrievals) agreed qualitatively on three main persistent melt events in terms of relative intensities and duration at CP1 during the melt season: a small one in late June, followed by the major melt

event that sustained whole July through late August, and a moderate one that begun in late August (on top of the sub-surface remnant melt), lasting through mid-September based on SMAP or early October based on the SEMB models (Figure 9a). For the early and late season melt events (light to moderate amount), the SEMB models estimated more LWA than any of the satellite retrievals with GEMB surpassing SAMIMI. However, during the peak melt event, we observed mixed results with the dielectric mixing models when comparing them to the SEMB models: the Colbeck and Looyenga models estimate higher LWA (Table 4) than the SEMB models, while the -Tiuri, Ulaby, Debye-like, Hallikainen, MätzlerMEMLS3, and Birchak models show lower estimates of LWA compared to the SEMB models (Table 4). These retrievals are lower than the ones Hossan et al. (2023) presented using the Ulaby model at this site during the same melt season. Wider Different constrains in frozen and melt season parameters, mainly density and background temperatures, explain some of these differences.

The Sihvola model was found to be in closest agreement with the SEMB models at CP1 until the peak of the melt season. Afterward, when active melting at the surface stops (as evidenced by gradual loss of LWA) and meltwater percolates deeper and refreezes, the satellite retrievals and the SEMB models exhibit more substantial differences (Table 6), which impacted the overall correlation and RMSD ( $0.74 \le r \le 0.84$  and  $18 \text{ mm} \le RMSD \le 28 \text{ mm}$ ; see Table 5 - 6). All the satellite retrievals consistently indicate a faster refreezing rate of the subsurface liquid water than in case of both the SEMB models, where the refreezing is determined by the evolution of the density profile and thermal conduction.

Figure 9: Comparison of the total daily liquid water amount (LWA) estimated from SMAP L-band TB observations with ten dielectric mixing models (solid lines), by the <u>SAMIMI</u> EBM (orange dash-dotted line) and GEMB (pink dotted line) forced with PROMICE and GC-Net AWS in situ measurements for Jun 1 – Oct 31, 2023.

The melt trends at DY2 (Fig. 9b) are similar to CP1 with some differences (Tables 3\_-\_6). The early season melt in late June is minor, while the late season peak is relatively higher with gradual loss of persistent subsurface melt that extended even beyond the end of September in consensus with the mixing models and with the SEMB models. Under such persistent liquid water and warmer subsurface background, while the surface recommences melting, liquid water is expected to infiltrate deeper and form a thicker wet layer. Satellite retrievals of average wet layer thickness by majority of the mixing models support this (Table 3). The overall agreement between the SEMB models and satellite retrieval with different mixing models is better at this site  $(0.956 \le r \le 0.98 \text{ and } 56 \text{ mm} \le \text{RMSD} \le 17 \text{ mm}$ ; see Tables 5-6).

KAN\_U is known to undergo extensive LWA throughout the summer (Hossan et al. 2023). However, all the mixing models, including SEMB models, report relatively lower LWA but prolonged melting conditions (Fig. 9c). Allmong the mixing models, except the MEMLS3 and Tinga model estimated higher average provide the thickness of the wet layer (-1 m) in close agreement with than the SEMB models (Table 3). Twhich is reasonable since the snowfall at this site is climatologically lower (MacFerrin et al., 2019; Machguth et al., 2016) and thicker ice layers underneath ought to prevent deeper infiltration. Regarding LWA, Tihe SEMB models showed better alignment with the Colbeck, Looyenga, and Sihvola model-based retrievals at the beginning of the peak melt (early July). However, as the melt season progresses and the snow/firn profile evolves, the differences intensify with the previous models, rather they better match with the rest of the models which claim lower estimates of LWA throughout. Firn models push the liquid water out of the system (called runoff) if the water balance exceeds certain limit (irreducible water saturation) determined by the available pore space. In reality, this excluded liquid water must still exist somewhere, which may explain some of these misalignments (Table 4 - 6). Nevertheless, the spread of maximum summer melts between the mixing models (satellite retrievals) is large (min 2147 mm by Tinga\_and Mätzler\_models to max 602 mm by Colbeck model with STD = 154 mm; see Table 43).

Since NSE is located at higher elevation (Table 1), historically it receives less frequent and less intense melt. In the 2023 melt season, we, however, observed similar melt trends with shorter duration compared to the previous three sites (Figure 9d). Only the GEMB model detected the early season melt in late June. Both GEMB and SAMIMI models estimate the presence of liquid water (max. 18 and 7 mm respectively) in the late melt season, which were completely ignoredalso detected by all the satellite retrievals (mixing models) despite enhanced emissions in the TB time series in Figure 8dthat better align with the estimate of SAMIMI model. This is because we used a spatially uniform thresholding technique (mean frozen season TB plus 10 STD; details in Sec. 2) that missed the detection. Compared to the pre-summer mean winter frozen TB used in the thresholding, the post-melt mean fallfrozen TB dropped around 5 K at this site due to-possibly from ice layers formed by refreezing of summer melt-exacerbating the falls negative problem, common in this higher elevation areas (Hossan et al 2024). Without dynamic thresholding, these late-season, less intense melt events would have been missed as false negatives. Other than this, although lesser in intensities, the order of magnitude of the satellite retrievals remains the same. However, the

results of the SAMIMI model better match with the group of mixing models that provide a lower estimate of LWA while the GEMB model provides the upper limit and better aligns with the models that indicate higher LWA. The GEMB model also refreezes noticeably more slowly than the SAMIMI model, refreezing faster and aligning better with the satellite retrievals.

Though all the satellite retrievals and SEMB models estimated an overall slightly higher LWA at SDL site in 2023, the trends closely replicated that of NSE (Figure 9e). The late season melt is however now stronger at this site, and the threshold algorithm detected melt and LWA was quantified by all the mixing models with the similar sequence – the Tinga model giving the lowest and the Colbeck model giving the highest maximum summer LWA (Table 4). Compared to the maximum summer LWA, the average thickness of the wet layer (Table 3) was found to be overall higher at this site, similar to NSE, which is anticipated since there is enough pore space but a lower number of ice layers allowing liquid water to percolate deeper. The SEMB models, despite differences between them, better align with the Sihvola, Looyenga, and Colbeck models than the rest of the others (Table 6), but both retain subsurface liquid water for elongated periods in the fall and-but indicate a thinner thickness of wet layer compared to the satellite retrieval with the majority of the mixing models (Table 3).

At the SDM site, only a single persistent melt event was observed; no significant melt was determined until the beginning of July; July, neither by the satellite retrievals nor by the SEMB models. The late season melt was also insignificant, only detected by SEMB models. We nevertheless see two minor spikes in the TB time series (Fig. 8), which were probably too small to be detected by the thresholding algorithm. The two SEMB models closely resemble at this site in phase and magnitude, but again they disagree with all mixing models in the satellite retrievals in the rate of subsurface refreezing in the late melt season, which impacted the comparison metrics in Table 5 and 6.

## 4 Discussion




The strong dielectric contrast between dry snow background and liquid water inclusion causes the complex dielectric constant of wet snow to significantly change with  $v_w \text{LWC [m^3-m^{-3}]}$ , which can vary over a wide range depending on the density (porosity) of the dry snow. Here, we focused on the GrIS percolation zone, where typical  $v_w \text{LWC}$  is known to be approximately 0.-.6 percent. Selected dielectric mixing models were found to vary widely over this narrow range, giving large uncertainties in modelling the effective depth of penetrations, TB, and consequently, in quantifying LWA based on the dielectric constant retrieved from satellite measurements. Differences of depolarization factors that describe the shape and orientation of the liquid water inclusion with respect to the emitting EM field mainly contribute to these differences for the structure dependent models. For the power law models, their degree controls the higher order local interactions in lieu of depolarization factor. There are significant uncertainties in the effective penetration depth, or emission contribution depths, between the models, especially when the  $v_w \text{LWC}$  (and hence the absorption) is low (pendular regime).

MWR TB is on the other hand a non-linear function of multiple parameters that gradually or abruptly vary with depth. Therefore, with limited knowledge of detailed snowpack properties and their evolution, modelling and interpretation of snow/firn microwave radiation, especially at L-band, which is sensitive from the surface to deeper layers, is difficult. Here we

used a simplified profile of temperature, density, and stratigraphy to simulate frozen and melt season TB at L-band. Nevertheless, LWA estimation with a single frequency is an underdetermined problem – even with constrained frozen background parameters, numerous combinations of volume fraction of melt and wet layer thickness can produce the same or close TB, but with different LWA. By using average measured dry snow density and parameterizing the wet layer thickness with an average retrieved thickness over melting days, we attempted to minimize some of the uncertainties.







The Debye-like, Hallikainen, and Ulaby models show higher TB sensitivity to lower  $v_w$  LWC at low density snow/firm background and provide lower estimates of overall LWA. The opposite results were observed when the background density is higher (see Fig. 4). This is counterintuitive because when the background density is low, there should be enough pore space in the snow to either hold more  $v_w$  LWC or to support deeper percolation or vice versa. The results of these models, however, align most closely with the empirical Tiuri model. This is encouraging in the sense that the Hallikainen and Ulaby models were originally derived and validated for measurements in the 3–37 GHz range and for a limited density range; these agreements support their applicability to L-band applications.

The Tinga model, on the other hand, shows more consistency with the change of dry snow density, yet it is highly nonlinear with  $v_w$  and provided the lowest LWA at five of the six AWS sites considered (except CP1, Fig. 4 and 59). The Mätzler MEMLS3 model provided an overall intermediate result in terms of effective dielectric constants (these results are in line with Picard et al. 2022 results), depth of penetration, TB, and LWA, and showed a reasonable fidelity over a wide range of density (Fig. 4). But compared to the SEMB models, which were forced by AWS measurements, Mätzler model MEMLS3, along with the above-mentioned models, provided the lowerst LWA in all six AWS (Figure 9 and Table 4).

So, if these models represent TB realistically, saturation at relatively low LWA would limit the liquid water estimation at L-band within a certain limit (approximately no more than 60 - 70 mm). The Colbeck model has a convincing theoretical and experimental basis, nevertheless this model consistently stayed apart from other models and provided the low end of the effective dielectric constant (and TB), and the high end of the LWA and penetration depth. The power-law dependent models, Birchak, Sihvola, and Looyenga, provided consistent estimates of the LWA and penetration depth throughout (dielectric constant and TB w.r.t.  $v_w$ LWC) in the order of lower to higher (higher to lower), respectively. Sihvola model was found to be the best match with SEMB models for the AWS and melt season considered (the overall RMSD at six AWS was ~ 11 mm; see Table 6). The advantage of these power law models is that they are easily configurable based on the degree of the model (a single parameter to fit -  $\beta$ ) and can easily be fit to the available ground truth (or SEMB estimates).

However, we refrain from recommending any particular model in this article, except exploring and demonstrating their individual and comparative characteristics under different  $v_w$ LWC, density, and other firn conditions, because caution should be taken when considering SEMB models as the reference for validating LWA estimates since they have their own limitations. Difference between the SEMB models when forced with the same inputs partly explains this. Consistent delayed refreezing in both the models (SAMIMI and GEMB) when melting at the upper surface ceases is an apparent indication of

inaccurate thermal conduction, affecting the overall LWA. Thermal conductivity of wet snow and wet-dry interface is probably lower. Future work should better parametrize these processes to refine the models.

Satellite derived LWA can be attributed to infiltrating water, occurring through two distinct modes of unsaturated flow: the downward propagation of a wetting front and the movement of water through preferential flow paths (also called pipes or flow fingers) (Marsh and Woo, 1984). Under sustained melting, wetting fronts typically form and propagate downward from the surface into the underlying cold firn (Colbeck, 1975), although their advancement can be hindered by nightly refreezing or by snowfall events. Additionally, structural features in the firn, such as ice layers or microstructure contrasts, can trap water. However, as water accumulates, it occasionally breaks through in highly heterogenous locations. The resulting preferential flow paths allow large volumes of water to infiltrate deep below the wetting front, bypassing the cold firn layers (Marsh and Woo, 1984); Pfeffer and Humphrey, 2022). The partitioning of meltwater between infiltration via wetting fronts versus preferential flow paths is highly variable and inherently difficult to predict due to its sensitivity to subtle firn structural and thermal conditions.




Several factors suggest that twice-daily LWA retrievals are more likely to reflect the water associated with the surface wetting front than liquid water contained within deeper preferential flow paths. First, the signal from water in subsurface pipes must propagate through the overlying wet layer, which has a stronger and more coherent L-band response due to its proximity to the surface and higher spatial continuity. Second, flow through preferential pathways is typically event-driven, with the breakthrough of accumulated water quickly penetrating deep into colder firn, where it often refreezes within hours rather than persisting for days or weeks (Humphrey et al., 2012). Finally, the L-band signal inherently averages over broad spatial footprints on the order of kilometres, favouring detection of the spatially extensive and homogeneous surface wet layer over the centimetre- to decimetre-scale, highly heterogeneous pipe structures. Therefore, future work should better understand and parametrize these processes to refine the algorithmmodel.

Parametrizations in the retrieval framework—such as the assumption of simplistic stratigraphy and liquid water distributions—may affect the absolute LWA estimates. However, these factors are likely to impact all models in a similar proportion. Therefore, the relative differences between the model estimates are more likely attributable to the specific formulations and assumptions of each model. Nevertheless, future work should aim to incorporate more advanced algorithms capable of resolving vertical profiles. Additionally Among others, a spatially and temporally dependent threshold should also be considered in future work to account for not only the ice layers due to refreezing but also the seasonal evolution of the snowpack that obviously contributed to the uncertainty in the results. The challenge would be the sensitivity and saturation of TB with increasing LWA. As shown, the signal power/field intensity decreases exponentially with depth and  $v_w$  LWC at a rate determined by the absorption and scattering coefficients (Fig. 3 and 7). Over absorption dominated regions, LWA estimates beyond  $\sim 60$ – $\sim 70$  mm would be highly uncertain. Future retrievals should consider a wider range of  $v_w$  LWC to incorporate the negative sensitivity (scattering dominated regions). To handle the inherent nonlinearities and dimensions of the problem,

advanced techniques (such as deep learning) may <u>be beneficial</u>help. Upcoming lower frequency missions (e.g., CryoRad: Macelloni et al., 2018) would also <u>create</u> offer new opportunities to sense deeper and enhance the capabilities.

# Conclusion

We investigated performances of ten dielectric mixing models for modelling wet snow TB at L-band to estimate the LWA in snow/firn column in the percolation zone of Greenland ice sheet. Six of the models (Tinga, Debye-like, Hallikainen, Ulaby, MätzlerMEMLS3, and Colbeck) are derivatives of fundamental mixing models (either MG or more generally, PVS) with empirically derived depolarization factors that account for the shape and orientation of the liquid water inclusions in dry snow background with respect to the emitting electromagnetic field. Except for the Colbeck model in this group, all models show overall relatively higher sensitivity of the effective dielectric constant to LWA, and thus TB, and generally produce lower estimates of LWA compared to the SEMB models. Colbeck model displays the lowest sensitivity of LWA to the effective dielectric constant and TB and, hence, yields the highest LWA of all models.

The differences among these models mostly originate from their depolarization factors that depend on multiple factors including density and LWA; they are deemed to be very difficult to quantify. Another group of models that follows power law relationships (Birchak, Sihvola, and Looyenga), not explicitly considering the depolarization factor, exhibit intermediate sensitivity of LWA to the effective dielectric constant and TB, and offer higher LWA than the former group of models (Tinga, Debye-like, Hallikainen, Ulaby, MätzlerMEMLS3, and Colbeck). A lower exponent (β) results in a lower sensitivity and a higher LWA, since a lower exponent allows the background to dominate in the mixing model. The results of the Tiuri model, which is fully based on empirical fitting to the field measurements at around 1 GHz, generally lie with that of the former group of models that explicitly consider the higher order interactions between the liquid water inclusions through depolarization factors. While the Hallikainen and Ulaby models were originally derived and verified for measurements made at 3 - 37 GHz, these agreements with Tiuri model support their applicability to L-band applications.

Compared to the SEMB models (SAMIMI and GEMB) driven by in situ observations, the first group of mixing models (Tinga, Debye-like, Hallikainen, Ulaby, Mätzler MEMLS3, and Tiuri) estimated consistently lower LWA in five of the six PROMICE GC-Net sites (except KAN\_U) which are more typical of percolation zone snow/firn physical conditions. Colbeck and Looyenga models measured consistently higher LWA than the SAMIMI and GEMB models in all six sites. In general, the Sihvola model aligned best with the SAMIMI and GEMB models for 2023 melt season. However, the best match does not imply correctness in the absence of an actual truth estimate. The SAMIMI and GEMB models disagree widely in certain cases; in general, the SEMB models have been found to produce diverging LWA estimates with the same in situ meteorological measurements foreing (e.g., Vandecrux et al., 2020; (Hossan et al., 2024; Moon et al., 2024)).

Despite the satellite retrievals using the different mixing models showing a wide variance in the total and maximum summer LWA, no significant discrepancies were observed in the timing of the onset and refreezing melt, which is based on the observed TB change. However, although satellite retrieval agrees well with the onset of melt with SEMB models,

significant disagreements were found in timing of complete refreezing of sub-surface liquid water in snow/firn. While all L-band retrievals indicate a sharper refreezing in all sites except KAN\_U, the SAMIMI and GEMB models seemed to refreeze slowly and retain sub-surface liquid water for an elongated period in post melt season. This was attributed to low thermal conductivity and slow heat transmission in the firn models. The differences between SAMIMI and GEMB models, even when they were run by the same set of in situ observations, are also indicative of the differences in their process representations.

This study sheds light on the behaviour of wet snow dielectric mixing models and consequent TB in presence of low liquid water ( $v_w$ : 0\_-\_6\_percent\_%, or LWA: 0\_-\_120 mm). The sensitivity and saturation behaviour of the models were broadly explored that gives an idea about the uncertainty associated with translating the L-band retrieved effective dielectric constant to LWA. Further work is required to better understand the melt water process in the snow and firn and their interactions with the microwave emissions. More sites specific in situ measurements of firn profiles under various conditions will be the next step to calibrate and validate these models to make better recommendation about using a group of specific mixing models.

# Data and code availability





SMAP Twice-Daily rSIR-Enhanced EASE-Grid 2.0 Brightness Temperatures, Version 2 data products were provided by National Snow and Ice Data Center and are publicly available at https://nsidc.org/data/nsidc-0738/versions/2. The PROMICE hourly AWS measurements are available at https://doi.org/10.22008/FK2/IW73UU (How et al., 2022). The **SUMup** subsurface profiles available temperature and density are at https://arcticdata.io/catalog/view/doi:10.18739/A2M61BR5M. SMAP and model LWA will be made available in a Zenodo repository. The scripts used to perform the analysis for this study will be shared through GitHub. MATLAB source code for glacier surface energy balance coupled with firn thermodynamic and hydrological modelling is available in PRISM Data: University of Calgary's Data Repository at https://doi.org/10.5683/SP2/WRWJAZ (Marshall, 2021).

## **Author contributions**

AH: concept development, method design, code implementation, formal analysis, results discussion, original draft, paper revision

AC: concept development, results discussion, original draft, paper revision, funding management, obtain funding

NS: GEMB outputs, results discussion, paper revision

JH: concept development, results discussion, paper revision

LA: results discussion, paper revision, obtain funding

JK: results discussion, paper revision, obtain funding

JM: results discussion, paper revision

RC: results discussion, paper revision, funding management, obtain funding

# **Competing interests**

The authors declare that they have no conflicts of interest.

# Acknowledgements

This work was funded by the NASA Cryospheric Sciences Program; The research was carried out at the Jet Propulsion

Laboratory, California Institute of Technology, under a contract with the National Aeronautics and Space Administration

(80NM0018D0004). © 2025. All rights reserved. We gratefully acknowledge computational resources and support from the

NASA Advanced Supercomputing Division. We thank Dr. Baptiste Vandecrux of Geological Survey of Denmark and

Greenland (GEUS) for his assistance in the sourcing and the accurately evaluating data from in situ measurements. We also
thank Prof Ghislain Picard for publicly sharing his SMRT model source codes. We thank Mary Jo Brodzik and -Molly Ann

Hardman of NSIDC/CIRES, University of Colorado Boulder for providing us SMAP enhanced resolution data products. The
Greenland maps were generated with the Arctic Mapping Tools (Greene et al., 2017). The first author benefited from
occasional use of ChatGPT (free version) for assistance with syntax and linguistic corrections during the preparation of this
manuscript.

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
