# Peer review of "Evaluation of Wet Snow Dielectric Mixing Models for L-Band Radiometric Measurement of Liquid Water Content in Greenland's Percolation Zone"

_EGUsphere, 2025_

## Referee Comment (RC2)

The paper provides a comprehensive evaluation of wet snow dielectric constant models and their application in L-band radiometry of liquid water content in Greenland. While I agree that it is of vital importance to have this type of evaluation, as many people underestimate the influence of the dielectric constant models, there are several issues needed to be resolved before it can be considered for publication:

1. I am concerned about your assumption for the middle-layer in the MEMLS - highly reflective firn layers. You assume that $\varepsilon_r$ varies from 5-26 while the imaginary part is fixed to 0.0002. This doesn't make much physical sense to me. For ice-firn mixture it is very unlikely to reach this high real part. If you assume some melting, then the imaginary part should also significantly increase. Please consider to either change the parameterization or justify this parameterization in the paper.

2. The paper has inconsistency in notations and some typos in equations:

- I could not find the definition of "v" in eq. (14.1) and "W" in eq. (16.1).

- Eqs. (17) and (18), β should be replaced by "1/2" and "0.4" respectively.

- The equation numbering is wrong, e.g, there are two eq. (8) and eq. (20) in the paper and eq. (15) has "(15)" and "(15.1)" while other equations start directly with (num.num), e.g. "(16.1)" and "(16.2)".

- eq. (20), there is often no "-" sign before k_0 in the definition of alpha.

3. I am not sure whether it is a good idea to have so many sections for different models - some sections only have 2-3 lines. I suggest either put all the models in one section, or the authors can group the models and put them in different sections, for instance, 2.3.1 - 2.3.3 can formulate a section named, for instance, "Debye-form models" and sections 2.3.8 – 2.3.10 can form a section named "Power-law models".

4. I recommend changing the name of the dielectric mixing model "MEMLS3" to "Matzler model" as the current name can cause some confusions to distinguish with the microwave emission model MEMLS3.

5. Notation consistency also needs to be improved.  Examples are given:

- I am particularly concerned about LWA, LWC and volume fraction of liquid water $v_w$. Is LWC the same as $v_w$? If so, please keep them the same everywhere in the paper. Furthermore,in Fig. 6, is the notation $m_v$ same as $v_w$?

- This also occurs in Fig. 4, does the notation twet mean $t_{wet}$? Please revise them.

- line 413, the percent is written in "percent" and later it is "%", please keep consistency.

6. The authors can consider using different colors to represent the values shown in the blocks for selected tables (e.g. Tables 3,4,5 and 6).  This colormap + number approach will greatly enhance the readability of these tables.

7. Caption of Fig. 9 – what is EBM? Perhaps you want to say SAMIMI?

8 . It is a bit surprise to see how big the impact of different dielectric mixing-model is on the retrieved LWA. Could it be associated with other parameterizations? For instance, the dielectric constant of middle layers? It would be good to briefly discuss this in the paper.

9. Line 716: please elaborate "forcing", e.g. weather and environmental conditions?

10.  Line 722: "except KAN_U…models seemed to refreeze" -> "except the retrievals using KAN_U…seemed to indicate the sites refreeze slowly…"

---

## Author Comment (AC4)

Authors' comment:

Dear Editor,

Thank you for your question. Mousavi et al. (2022) modeled the dielectric constants of medium 2 (wet snow) and medium 4 (semi-infinite dry snow) in a four-layer configuration using the formulations of Ulaby and Long (2014, Chapter 4, pp. 140–145, Eq. 4.55 – 4.61). However, they explicitly prescribed the complex dielectric constant of medium 3 (the highly reflective layer) as 3.5 – 9j. For details, please refer to Table II in Mousavi et al. (2022), where the layer properties of the four-layer model are listed (corresponding to the Fig. 3b where the semi-infinite air medium was considered as a separate layer).

While we follow a similar modeling approach, we use different values for the complex dielectric constant of the reflective layer ($\varepsilon_2 = \varepsilon_r - 0.0002$j), as we consider such high absorption (caused by loss factor 9j) to be unlikely in dry snow. Instead, we hypothesize that successive reflection caused by complex stratigraphy is the dominant mechanism. Therefore, we tune the real part of $\varepsilon_2$ to match the simulated and observed brightness temperatures at each grid point during the frozen season.

Sincerely,

Alamgir Hossan (on behalf of the authors)

---

## Author Response (AR1)

**Dear Editor,**

Thank you for informing us of your decision and for facilitating the review process. We appreciate the reviewers' constructive feedback and have revised the manuscript accordingly. Below, we address each comment point by point. In addition, following your earlier recommendation, we have updated the color scheme of all the figures to ensure the revised manuscript is accessible to readers with color vision deficiencies.

Best regards,

Alamgir Hossan (on behalf of all the authors)

Author Response to Reviewer #1 (Reviewer's comments are in black and author responses are highlighted in blue).

Overall Review: The article provides a comparison between 10 different wet-snow dielectric mixing models and how the choice can influence the retrievals of liquid water content at L-band frequency. The study was conducted for Greenland's percolation zone, utilizing SMAP (rSIR) brightness temperature and two in-situ forced surface-energy mass balance models to assess robustness. The combination of models and satellite observations is unique and novel and answers an important question about the choice of wet-snow dielectric models for the estimation of liquid water amount. The manuscript is generally well organized and richly referenced, but a careful read reveals several minor errors that should be addressed before publication.

**Major Strengths & Novelty**

- 1. First side-by-side comprehensive comparison of ten-dielectric formulations (Debye-like, power-law, empirical, etc.) explicitly focused on the retrieval of LWC.
- 2. Link to operational satellite SMAP rSIR Tbs
- 3. Quantitative assessments against the surface energy models.
- The authors thank the reviewer for careful review and thoughtful comments. Below is our point-by-point response to the reviewer's 'Specific Questions and Technical Errors'.

- 1. Abstract "Sihvola power-law mixing model showed an overall better performance than the other models for the 2023 melt season" consider including metrics.
  - We revised the abstract with quantitative metrics in the revised manuscript (lines 31 36 in the marked-up manuscript version).
- 2. Which SMAP product was used (rSIR) can be mentioned in the introduction, in the last paragraph.
  - The SMAP data product used (rSIR enhanced resolution) has been included in the last paragraph of the introduction.
- 3. Duplicate equation number (for eq. 8 mentioned at L137 and L155), and then subsequent equation numbers should be changed.
  - Thanks for noticing this. Duplications for equation numbers have been corrected for the equations as well as in the text throughout the revised manuscript.
- 4. Typo in L273 Ks<<Ks, instead of Ks<<Ka.
  - The typo has been corrected in the revised manuscript.
- 5. Methods The Hallikainen model was derived at 3-37 GHz; authors can justify its usage/extrapolation to 1.4GHz
  - While the Hallikainen model was derived using measurements made at 3-37 GHz, our purpose was to test its application at 1.4 GHz to see how it performs against the other models as mentioned in the manuscript. Considering the results, the agreements of the model were very close to the Tiuri model which was derived from in situ measurements made at 859 MHz 12.6 GHz. It supports Hallikainen model for its extended applicability to L-band. We included this justification in the discussion section (lines 812 813 in the marked-up manuscript version).
  - •
- 6. Typo Table 1: Key Parameters "Depolarization" should be "Depolarization".
  - The typo has been corrected.
- 7. Sihvola, misspelled at L99, L123, L166 as Sihivola.
  - The typos have been corrected.
- 8. Eq 20 refers to both e-folding depth and attenuation coefficient.

- The duplications for Equations 20 have been corrected.
- 9. Hallikainen et. al. 1984 (L344) is not mentioned in the bibliography; are the authors referring to Hallikainen et. al. 1986? If so, the date should be changed.
  - Yes, we referred to Hallikainen et. al. (1986). The citation has been corrected in the revised manuscript.
- 10. I suggest making the zoomed-in version on the right in Fig. 2
  - We have changed the zoomed-in version to the right in Fig. 2, consistent with Fig. 3, in the revised manuscript.
- 11. L427 "The Colbeck model provides the lowest estimates for the entire LWA range," referencing Fig. 4. However, in Fig. 4f, the Colbeck model appears to provide a higher estimate than Hallikainen. (A zoomed-in inset for Figs. 4, and 5,6 would be helpful).
  - We tried to highlight the general trends for most of the cases. But it's true that the Colbeck model appears to provide a higher estimate than Hallikainen in Fig. 4f. The statement has been revised accordingly (line 504 in the marked-up manuscript version).
- 12. Table 3 GEMB column is missing.
  - GEMB results have been included in Table 3 of the revised manuscript.
- 13. Line 550, "All three methods...", is it referring to Fig.9?
  - Yes, it refers to Figure 9. We explicitly included the figure number at the end of the line.
- 14. Can include a plot of observed and simulated tb, to check the loss.
  - We have included simulated TBs with the observed TBs given in Figure 8, as recommended. Relevant texts were also revised accordingly (lines 578 581 in the marked-up manuscript version).

.

- 15. Line 885 (+more) Miller, J.Z. has year 2020a, but 2020b is missing, I see that at Line 897 Miller, O., et. al, has the year 2020b. but no corresponding 2020a.
  - The references were previously processed incorrectly and have now been corrected as follows: 'Miller, J.Z. 2020a' has been changed to 'Miller, J.Z. (2020)', and 'Miller, O., et al. (2020b)' has been changed to 'Miller, O., et al. (2020).

Author Response to Reviewer #2 (Reviewer's comments are in black and author responses are highlighted in blue).

The paper provides a comprehensive evaluation of wet snow dielectric constant models and their application in L-band radiometry of liquid water content in Greenland. While I agree that it is of vital importance to have this type of evaluation, as many people underestimate the influence of the dielectric constant models, there are several issues needed to be resolved before it can be considered for publication:

- The authors thank the reviewer for comprehensive review and insightful comments. Below is our point-by-point response to the reviewer's comments.
- 1. I am concerned about your assumption for the middle-layer in the MEMLS highly reflective firn layers. You assume that  $\epsilon_r$  varies from 5-26 while the imaginary part is fixed to 0.0002. This doesn't make much physical sense to me. For ice-firn mixture it is very unlikely to reach this high real part. If you assume some melting, then the imaginary part should also significantly increase. Please consider to either change the parameterization or justify this parameterization in the paper.
  - The near-surface density profile in the percolation zone is highly variable and characterized by multi-scale fluctuations (Johnson et al., 2014 and the references therein). Moreover, seasonal melting and refreezing lead to the formation of complex features such as random ice layers and ice pipes. Accurately modeling these effects across the percolation zone remains a significant challenge due to the lack of detailed ancillary data (e.g., temperature, density) and is the subject of ongoing research. These sub-grid-scale structural variabilities contribute to the significant scattering of L-band brightness temperatures, particularly during the frozen season (Hossan et al., 2024). To account for these effects without introducing multiple uncertain parameters, we chose to model the combined reflective impact of the complex firn stratigraphy using an equivalent dielectric slab with a tuned permittivity (real part), following an approach similar to Mousavi et al., (2022).

This equivalent layer is located beneath the seasonal dry/wet snowpack (top layer) and is defined by a real permittivity value that varies spatially (at each grid point) but remains constant temporally throughout the year. We acknowledge that, for typical ice-firn mixtures, such high values of the real part may seem unrealistic; however, it is important to note that this layer does not contain liquid water; its purpose is to simulate equivalent dielectric contrasts (for combined reflectivity) beneath the seasonal dry/wet snow rather than to represent physical dry or wet snow structures. Therefore, we maintain a low and fixed value for the imaginary part of the permittivity (0.0002), consistent with dry snow or ice. For a fixed location, the same characteristics of this layer were applied to all the models. As such, this layer has a negligible impact on liquid

- water retrievals, which are governed by the top-layer that explicitly accounts for varying water volume fraction and thickness.
- We have included this explanation to the revised manuscript (lines 375 393 in the marked-up manuscript version) to justify our parameterization choice as recommended.
- 2. The paper has inconsistency in notations and some typos in equations:
- I could not find the definition of "v" in eq. (14.1) and "W" in eq. (16.1).
  - "v" in Eq. (14.1) was meant to represent the volume fraction of liquid water ( $v_w$ ). But we notice it was mixed with "W" in Eq. (16.1), and with f at some other places (Eq. 17-19). We replaced it with a single parameter ( $v_w$ ) to represent the volume fraction of liquid water throughout the revised manuscript.
- Eqs. (17) and (18),  $\beta$  should be replaced by "1/2" and "0.4" respectively.
  - As a part of the reorganization of Sec. 2.3 (as per review comment 3), we have removed the equations. For all the power law-based models, we now refer to Eq. 8, with respective values for the exponent (β), also summarized in Table 1.
- The equation numbering is wrong, e.g, there are two eq. (8) and eq. (20) in the paper and eq. (15) has "(15)" and "(15.1)" while other equations start directly with (num.num), e.g. "(16.1)" and "(16.2)".
  - The numbering of the equations has been corrected, and format has been ensured consistent.
- eq. (20), there is often no "-" sign before k 0 in the definition of alpha.
  - We agree some authors do not follow the negative sign convention for the attenuation coefficient. Here, we followed the Ulaby and Long, (2014) convention to ensure the attenuation constant α is positive, given that the imaginary part of the complex square root can be negative in a lossy medium.
- 3. I am not sure whether it is a good idea to have so many sections for different models some sections only have 2-3 lines. I suggest either put all the models in one section, or the authors can group the models and put them in different sections, for instance, 2.3.1 2.3.3 can formulate a section named, for instance, "Debye-form models" and sections 2.3.8 2.3.10 can form a section named "Power-law models".
  - As recommended, we reorganized and thoroughly revised the subsection (Sec. 2.3) in the revised manuscript. All the wet snow models are included in one section, and the order follows the same as the models in Table 1. We also revised the background section (Sec. 2.1) to improve logical order and readability.

- 4. I recommend changing the name of the dielectric mixing model "MEMLS3" to "Matzler model" as the current name can cause some confusions to distinguish with the microwave emission model MEMLS3.
  - We have changed the name of the dielectric mixing model "MEMLS3" to "Matzler model" in the revised manuscript as recommended.
  - We have also changed the order of the models in the plots and tables to make it consistent throughout the manuscript.
- 5. Notation consistency also needs to be improved. Examples are given:
- I am particularly concerned about LWA, LWC and volume fraction of liquid water vw. Is LWC the same as vw? If so, please keep them the same everywhere in the paper. Furthermore, in Fig. 6, is the notation mv same as vw?
  - The liquid water content (LWC) and volume fraction of liquid water  $(v_w)$  are the same, both indicating volumetric liquid water inclusion in percent. On the other hand, liquid water amount (LWA) is the product of  $v_w$  and the thickness of the wet layer  $t_{wet}$  and expressed in m. w. e unit (defined in Eq. 20 in the revised manuscript). In the revised manuscript, we merged LWC/ $v_w$  into a single parameter  $v_w$  to avoid confusion and make it consistent throughout the manuscript.
  - In Fig. 6,  $m_v$  has been replaced by  $v_w$  as well.
- This also occurs in Fig. 4, does the notation twet mean twet? Please revise them.
  - Yes, "twet" was used to mean  $t_{wet}$  in Fig. 4. It is corrected as  $t_{wet}$ .
- line 413, the percent is written in "percent" and later it is "%", please keep consistency.
  - All "%"s has been replaced with "percent" in the revised manuscript.
- 6. The authors can consider using different colors to represent the values shown in the blocks for selected tables (e.g. Tables 3,4,5 and 6). This colormap + number approach will greatly enhance the readability of these tables.
  - We experimented with incorporating different colors to represent the values in these tables, as recommended. However, we felt that extensive use of colors might distract from the clarity and simplicity of the presentation. Therefore, we opted to keep the color variations minimal (black and white), while still ensuring the tables remain clear and informative.
- 7. Caption of Fig. 9 what is EBM? Perhaps you want to say SAMIMI?
  - Yes, we referred to the SAMIMI energy balance mode by 'EBM'. In the revised manuscript, we added SAMIMI before it (SAMIMI EBM).

- 8. It is a bit surprise to see how big the impact of different dielectric mixing-model is on the retrieved LWA. Could it be associated with other parameterizations? For instance, the dielectric constant of middle layers? It would be good to briefly discuss this in the paper.
  - The choice of dielectric models significantly impacts the LWA retrieval as the manuscript concludes. It can be associated with the other parametrizations, but the dielectric constant of the middle layer has little impact as mentioned in response 1. One of the crucial factors was the density of dry snow background. To minimize the uncertainty from density, we used the average measured density from the top 3 meters of snow, and it was fixed for all the models for a particular AWS. Other retrieval issues such as assumption of simplistic stratigraphy, and liquid water distributions may affect the absolute LWA estimates, and it should impact all the models in similar proportion. But we believe the relative differences between the estimates come from the respective model formulations and their assumptions.
  - Nevertheless, we have further revised the results and methodology. Specifically, the following changes were made:
    - For the models using dry snow as a background (Mätzler and power law-based models), we ensured the use of the Mätzler (2006) model (Eqs. 10–11).
    - To account for the difference between pre- and post-summer mean frozenseason TB—potentially caused by crust formation due to refreezing in higher elevation areas—we revised the threshold algorithm to use separate frozen references for these two cases. This improved the overall performance of the algorithm, enabling the detection of the late-season mild melt events that were previously mis-detected with the earlier version (Fig. 9). These updates are reflected in Tables 3 – 6 and Figures 8 – 9.
    - We also revised the discussion section to clarify the changes and consider other potential contributing factors.
- 9. Line 716: please elaborate "forcing", e.g. weather and environmental conditions?
  - The 'forcing' in line 716 indicated the in situ meteorological measurements from the automatic weather stations (AWS) (air temperature, air pressure, upwelling and downwelling short and longwave radiation fluxes, snow-surface height, wind speeds). To clarify, we replaced 'forcing' with 'in situ meteorological measurements (line 821 in new marked up version).
- 10. Line 722: "except KAN\_U...models seemed to refreeze" -> "except the retrievals using KAN\_U...seemed to indicate the sites refreeze slowly..."
  - Except for the average density of 3 m, the retrievals are independent of any site-specific in-situ data. We intended to indicate the retrieval (the SMAP measurement indirectly) at KAN U site.

**References:**

Hossan, A., Colliander, A., Vandecrux, B., Schlegel, N.-J., Harper, J., Marshall, S., and Miller, J. Z.: Retrieval and Validation of Total Seasonal Liquid Water Amounts in the Percolation

Zone of Greenland Ice Sheet Using L-band Radiometry, EGUsphere, 2024, 1–33, https://doi.org/10.5194/egusphere-2024-2563, 2024.

Johnson, J. T., Jezek, K. C., and Tsang, L.: UWBRAD: Ultra-Wideband Software-Defined Microwave Radiometer for Ice Sheet Subsurface Temperature Sensing, 2014.

Mousavi, M., Colliander, A., Miller, J., and Kimball, J. S.: A Novel Approach to Map the Intensity of Surface Melting on the Antarctica Ice Sheet Using SMAP L-Band Microwave Radiometry, IEEE J. Sel. Top. Appl. Earth Obs. Remote Sens., 15, 1724–1743, https://doi.org/10.1109/JSTARS.2022.3147430, 2022.

Ulaby, F. and Long, D.: Microwave Radar and Radiometric Remote Sensing, Microw. Radar Radiom. Remote Sens., https://doi.org/10.3998/0472119356, 2014.

---

## Author Response (AR2)

**Dear Editor.**

Thank you for accepting the manuscript for the publication. We have revised the manuscript to address the reviewer #1 request to include a clearer justification of the middle layer parameterization, the rationale for tuning the dielectric constant, and a complete citation for Ulaby and Long (2014). Please, see the marked-up manuscript version.

We appreciate your coordination to complete the review process in time.

Best regards,

Alamgir Hossan (on behalf of all the authors)

**Specific Revisions:**

- 1. Clearer justification of the middle layer parameterization: We revised lines 317-317 of the revised manuscript (marked-up version) to make the justification clearer.
- 2. The rationale for tuning the dielectric constant: The rationale for tuning the dielectric constant of the middle layer is to determine the baseline emission during frozen season as described in lines 291-304. This is done by matching the simulated TB with the observed TB during frozen season. Lines 292 and 344 in the revised manuscript (marked-up version) have been revised to clarify this rationale.
- 3. Complete citation for Ulaby and Long (2014): The citation has been edited (page number added) throughout the revised manuscript. The reference has also been revised, and other minor syntaxes have been corrected as well.

---

## Author Response (AR3)

**Evaluation of Wet Snow Dielectric Mixing Models for L-Band Radiometric Measurement of Liquid Water Content in Greenland's Percolation Zone**

- 5 Alamgir Hossan1, Andreas Colliander1, Nicole-Jeanne Schlegel2, Joel Harper3, Lauren Andrews4, Jana Kolassa4,5, Julie Z Miller6,7, Richard Cullather4,87
  - 1Jet Propulsion Laboratory, California Institute of Technology, Pasadena, California, United States
  - 2NOAA/OAR Geophysical Fluid Dynamics Laboratory (GFDL), Princeton, New Jersey, United States
- 10 3Department of Geosciences, University of Montana, Missoula, Montana, United States
  - 4NASA Global Modeling and Assimilation Office, Goddard Space Flight Center (GSFC), GSFC/GMAO-Greenbelt, Maryland, United States<del>MD</del>
  - 5Science Systems and Applications (SSAI), Berwyn Heights, Maryland, United States MD Department of Geography, University of Calgary
- 6EarthSAR, LLC, Salt Lake City, Utah, United States

20

- 76Cooperative Institute for Research in Environmental Sciences, University of Colorado Boulder, Boulder, Colorado, United States
- 87 UEarth System Science Interdisciplinary Center (ESSIC), University of Maryland at College Park, Maryland, United Statesniv. of Maryland at College Park, ESSIC, College Park, MD

Correspondence to: Alamgir Hossan (alamgir.hossan@jpl.nasa.gov) and Andreas Colliander (andreas.colliander@jpl.nasa.gov)

Abstract. The effective permittivity of wet snow and firn links the snow microphysics to its radiometric signature, making it essential for accurately estimating the liquid water amount (LWA) in the snowpack. Here, we compare ten commonly used microwave dielectric mixing models for estimating LWA in wet snow and firn using L-band radiometry. We specifically focus on the percolation zone of the Greenland Ice Sheet (GrIS), where the average volume fraction of liquid water is between 0 and 6 percent. We used L-band brightness temperature (TB) observations from the NASA Soil Moisture Active Passive (SMAP) mission in an inversion-based framework to estimate LWA, applying different dielectric mixing formulations in the forward simulation. We compared the effective permittivities of the mixing models over a range of conditions and evaluated their impact on the LWA retrieval. We also compared the LWA retrievals to the corresponding LWA from two state-of-the-art Surface Energy and Mass Balance (SEMB) models. Both SEMB models were forced with in situ measurements from automatic weather stations (AWS) of the Programme for Monitoring of the Greenland Ice Sheet (PROMICE) and Greenland Climate Network (GC-Net) located in the percolation zone of the GrIS and initialized with relevant in situ profiles of density, stratigraphy, and sub-surface temperature measurements. The results show that the mixing models produce substantially different real and imaginary parts of the dielectric constant, significantly impacting the LWA retrieved from the TB. The

correspondence with the SEMB-derived LWA varied by model and site, with correlation coefficients ranging from 0.67 to 0.98 and RMSD values between 5.4 and 23.9 mm. Overall, the power law-based empirical models demonstrated better performance for 2023 melt season. The analysis supports informed selection of dielectric mixing models for improved LWA retrieval accuracy.

[revised manuscript text omitted]
 using MEMLS V3 (Mätzler and Wiesmann, 2012). In this configuration (Figure 1), the top layer-represents the seasonal snowpack,-dry snow/firn during frozen season and wet snow during melt. The bottom layer of the three-layer configuration is treated as semi-infinite ice. To account for the influence of internal firn stratigraphy, we introduced a middle layer modelled as an effective dielectric slab. This layer captures the bulk reflective effects of complex subsurface layering in background emission, following a similar approach used by Mousavi et al. (2022). Rather than representing any specific physical layer, it serves to model the cumulative 
[revised manuscript text omitted]